# Milestone-Guided Policy Learning for Long-Horizon Language Agents

**Zixuan Wang** [1 2]   **Yuchen Yan** [1]   **Hongxing Li** [1]   **Teng Pan** [1 2]   **Dingming Li** [1]   **Ruiqing Zhang** [2]   **Weiming Lu** [1]
**Jun Xiao** [1]   **Yueting Zhuang** [1]   **Yongliang Shen** [† 1]

## Abstract

While long-horizon agentic tasks require language agents to perform dozens of sequential decisions, training such agents with reinforcement learning remains challenging. We identify two root causes: credit misattribution, where correct early actions are penalized due to terminal failures, and sample inefficiency, where scarce successful trajectories result in near-total loss of learning signal. We introduce a milestone-guided policy learning framework, **BEACON**, that leverages the compositional structure of long-horizon tasks to ensure precise credit assignment. BEACON partitions trajectories at milestone boundaries, applies temporal reward shaping within segments to credit partial progress, and estimates advantages at dual scales to prevent distant failures from corrupting the evaluation of local actions. On ALFWorld, WebShop, and ScienceWorld, BEACON consistently outperforms GRPO and GiGPO. Notably, on long-horizon ALFWorld tasks, BEACON achieves 92.9% success rate, nearly doubling GRPO's 53.5%, while improving effective sample utilization from 23.7% to 82.0%. These results establish milestone-anchored credit assignment as an effective paradigm for training long-horizon language agents. Code is available at https://github.com/ZJU-REAL/BEACON.

## 1. Introduction

Large language model agents have demonstrated remarkable capabilities in performing complex tasks in diverse environments (Yao et al., 2023; Schick et al., 2023), including web navigation (Zhou et al., 2024a; Deng et al., 2023), embodied control (Ahn et al., 2022; Huang et al., 2023; Wang et al.,

2026), and scientific experimentation (Boiko et al., 2023; Bran et al., 2023). These agents must perform sequences of decisions that span dozens of steps, with success determined only at task completion. Training such agents through reinforcement learning has shown promise (Zhang et al., 2026a; Ouyang et al., 2022), yet current policy optimization methods scale poorly with task horizon, exhibiting systematic performance collapse as decision sequences lengthen.

This collapse stems from two fundamental limitations of trajectory-level optimization, which treats trajectories as flat action sequences and assigns credit based solely on terminal outcomes. The first is *credit misattribution*: all actions within a trajectory receive identical advantages based solely on the terminal outcome. A correct early action is penalized when later actions cause failure; the same action receives opposite gradient signals across trajectories depending on downstream stochasticity, causing gradients to conflict. The second is *sample inefficiency*: as task horizons extend, successful trajectories become increasingly scarce, causing most samples to yield zero reward. Moreover, trajectories that complete substantial subgoals but fail the final objective receive zero reward identical to complete failures, wasting meaningful progress. We validate these limitations on ALFWorld (Shridhar et al., 2021): GRPO (Shao et al., 2024) achieves 77% success on short tasks but collapses to 54% on long tasks, with over 40% of gradient updates containing contradictory signals. Furthermore, 39% of sampled trajectories complete at least one subgoal yet contribute no learning signal under trajectory-level optimization.

Existing methods that aim to provide denser credit assignment introduce their own limitations. Process reward models (Lightman et al., 2024; Wang et al., 2024) require expensive step-level annotations and risk reward hacking (Gao et al., 2022). Monte Carlo value estimation (Kazemnejad et al., 2025) demands multiple rollouts per decision point, multiplying computational cost. GiGPO (Feng et al., 2025) constructs step-level comparison groups by identifying repeated states across trajectories, but its effectiveness depends on state recurrence, which diminishes as agents progress toward task completion in long-horizon settings. We observe that long-horizon agentic tasks already exhibit exploitable structure: they decompose into phases bounded by *milestones*, state transitions where subgoal achievement

---

[†]Corresponding author. [1]Zhejiang University [2]Baidu Inc.. Correspondence to: Yongliang Shen <syl@zju.edu.cn>.

*Proceedings of the 43rd International Conference on Machine Learning*, Seoul, South Korea. PMLR 306, 2026. Copyright 2026 by the author(s).

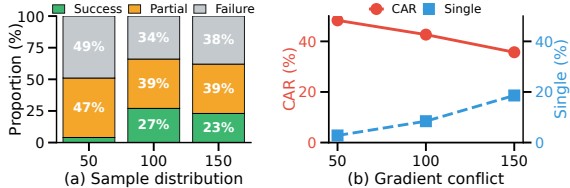

*Figure 1.* **BEACON overview and performance preview. Left:** GRPO assigns uniform credit from terminal outcomes, penalizing correct early actions when later actions fail; BEACON partitions trajectories at milestones and estimates advantages at dual scales. **Right:** On ALFWorld, GRPO degrades sharply with task horizon while BEACON maintains robust performance across all horizons.

renders prior execution history largely irrelevant. This approximate Markov property enables credit to be decoupled across phases, yet trajectory-level methods ignore it entirely.

We introduce Milestone-Guided Policy Learning Framework (BEACON), which leverages task structure to address both credit misattribution and sample inefficiency. The key idea is to partition trajectories at milestone boundaries and perform credit assignment at the segment level rather than the trajectory level. Given a trajectory, BEACON first identifies milestones from verifiable state changes, and partitions the trajectory into segments accordingly. Within each segment, temporal reward shaping assigns higher credit to actions closer to milestone completion, transforming sparse terminal signals into dense feedback that rewards partial progress. Across segments, dual-scale advantage estimation computes advantages at both trajectory and segment levels. The trajectory-level advantage captures global task performance, while the segment-level advantage compares only among trajectories that reached the same milestone, isolating local action quality from the variance introduced by subsequent segments. This decomposition ensures that a correct action in an early segment is not penalized by failures in later segments, directly addressing credit misattribution.

We evaluate BEACON on ALFWorld (Shridhar et al., 2021), WebShop (Yao et al., 2022), and ScienceWorld (Wang et al., 2022). BEACON outperforms GRPO across all benchmarks, with improvements that amplify as task horizons extend: relative gains over GRPO scale from 26.2% on short tasks to 73.6% on long tasks on ALFWorld. On Long tasks, BEACON achieves 92.9% success versus 53.5% for GRPO. Analysis reveals that BEACON recovers learning signal from partial successes: effective sample utilization improves from 23.7% to 82.0%. Furthermore, BEACON achieves 91.4% success compared to 43% for supervised fine-tuning on oracle trajectories, confirming that the gains stem from policy optimization rather than milestone imitation.

In summary, our contributions are as follows:

- This work identifies credit misattribution and sample in-

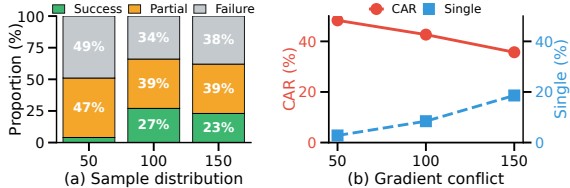

*Figure 2.* **Failures in flat trajectory optimization. (a)** Sample distribution during GRPO training. Partial successes yield zero gradient despite meaningful progress. **(b)** Gradient conflict analysis. Contradictory signals cause effective learning signal to collapse.

efficiency as fundamental limitations of trajectory-level optimization, showing that over 40% of gradient updates contain contradictory signals as task horizons extend.

- We propose BEACON, a framework that partitions trajectories at milestone boundaries, applies temporal reward shaping within segments, and estimates advantages at dual scales to isolate local action quality from later failures.

- Experiments on ALFWorld, WebShop, and ScienceWorld demonstrate horizon-dependent improvements, with relative gains over GRPO scaling from 26.2% to 73.6% and sample utilization improving from 23.7% to 82.0%.

## 2. Failures in Flat Trajectory Optimization

We first establish empirically that trajectory-level policy optimization fails systematically as task horizons extend, then diagnose the underlying causes through gradient analysis.

Experiments use Qwen2.5-1.5B-Instruct (Qwen et al., 2025) on ALFWorld with GRPO, stratifying tasks by optimal trajectory length: Short ($L^* \leq 4$), Medium ($5 \leq L^* \leq 7$), and Long ($L^* > 7$) (details in Section 4.1). Figure 1(Right) shows GRPO degrades from 76.7% (Short) to 53.5% (Long).

**Sample Inefficiency.** Figure 2(a) shows the sampled trajectory distribution during training. We categorize trajectories into three types: full successes (green) that complete the task, partial successes (orange) that complete at least

one milestone but fail the final task, and complete failures (gray) that achieving none. Partial successes consistently comprise 39–47% of samples throughout training, yet under GRPO they receive zero reward identical to complete failures. Meanwhile, full successes remain below 27%, meaning over 73% of samples yield no learning signal. This waste of partial progress severely limits learning efficiency.

**Credit Misattribution.** Even among trajectories that do provide signal, credit assignment is corrupted. Figure 2(b) reveals a second pathology: gradient corruption from contradictory credit assignment. We measure the Contradictory Action Ratio (CAR), defined as the fraction of actions that receive opposite-sign advantages across different trajectories despite being executed at identical states. CAR exceeds 40% at its peak, indicating that nearly half of gradient updates for repeated state-action pairs point in conflicting directions. As a consequence, the effective learning signal (the fraction of gradient that survives after cancellation) collapses below 20% (see Appendix C.2 for detailed computation). The root cause is that trajectory-level advantages conflate action quality with downstream stochasticity: the same correct action receives positive credit when later actions succeed and negative credit when they fail.

**Takeaways.** Flat trajectory optimization suffers from two compounding problems. Sample inefficiency discards learning signal from partial successes, while credit misattribution corrupts the signal that remains. Both problems worsen as horizons extend: longer tasks have lower success rates (increasing partial successes) and more opportunities for downstream variance to corrupt credit assignment. Addressing these failures requires exploiting the compositional structure that trajectory-level methods ignore.

## 3. Milestone-Anchored Policy Optimization

We introduce BEACON, a framework that exploits the compositional structure of long-horizon tasks to address the credit assignment failures identified in Section 2. BEACON operates in three stages: partitioning trajectories at milestone boundaries, shaping rewards within segments, and estimating advantages at dual scales.

### 3.1. Preliminaries

We consider a Markov Decision Process $(\mathcal{S}, \mathcal{A}, \mathcal{P}, \mathcal{R}, \gamma)$ where a language agent policy $\pi_\theta$ produces trajectories $\tau = \{(s_t, a_t)\}_{t=1}^T$ through interaction with an environment. The agent receives sparse terminal reward $R(\tau) \in \{0, 1\}$ indicating task success.

We assume access to a milestone indicator $\Phi : \mathcal{S} \times \mathcal{A} \times \mathcal{S} \rightarrow \{0, 1\}$ that returns 1 when a transition completes a semantic subgoal, and 0 otherwise. Crucially, $\Phi$ does not require

learned models or manual annotation—it detects observable state changes from environment feedback. In interactive environments, such signals are typically available: in ALF-World, $\Phi$ detects object state changes such as successful pick-up or heating completion; in WebShop, $\Phi$ identifies page transitions advancing toward the target product; in ScienceWorld, the environment provides explicit subgoal signals that $\Phi$ directly consumes.

### 3.2. Trajectory Partitioning

Long-horizon tasks naturally decompose into phases bounded by milestone states. Given trajectory $\tau$, applying $\Phi$ to each transition yields milestone timestamps $\mathcal{M} = \{t_1, \ldots, t_K\}$ where $K$ is the number of milestones reached. Setting $t_0 = 0$ and $t_{K+1} = T$, we partition $\tau$ into $K + 1$ segments:

$$\text{Seg}_k = \{(s_t, a_t) : t_{k-1} < t \le t_k\}, \quad k \in \{1, \ldots, K+1\}. \tag{1}$$

We partition at milestone boundaries based on the following structural assumption:

**Assumption 3.1** (Milestone Markov Property). For milestone state $s_{t_k}$ reached at timestep $t_k$:

$$\begin{aligned} P(\text{Seg}_{k+1}, \ldots, &\text{Seg}_{K+1} \mid s_{t_k}, \text{Seg}_1, \ldots, \text{Seg}_k) \\ &\approx P(\text{Seg}_{k+1}, \ldots, \text{Seg}_{K+1} \mid s_{t_k}). \end{aligned} \tag{2}$$

This assumption states that conditioned on reaching a milestone state, future trajectory distribution depends primarily on remaining subgoals rather than the full history. This is natural for compositional tasks: once an object is picked up, subsequent success depends on what to do next, not on how the object was found. We discuss the validity and limitations of this assumption in Appendix A.2.

### 3.3. Temporal Reward Shaping

Partitioning alone does not address sample inefficiency, since segments in failed trajectories still receive zero reward. We assign shaped rewards crediting partial progress.

For action $a_t$ in segment $\text{Seg}_k$ of trajectory $\tau_i$ with $K_i$ completed milestones:

$$r_t = \begin{cases} R_{\text{ms}} \cdot \gamma^{t_k - t} & \text{if } k \le K_i \\ 0 & \text{if } k = K_i + 1 \end{cases}, \tag{3}$$

where $R_{\text{ms}} > 0$ is the milestone reward and $\gamma \in (0, 1)$ is the temporal decay factor. Only segments that end with a completed milestone receive positive reward. This design has two properties: (1) all actions in completed segments receive positive reward, enabling learning from partial successes; (2) actions closer to milestone completion receive higher credit, encouraging efficient execution.

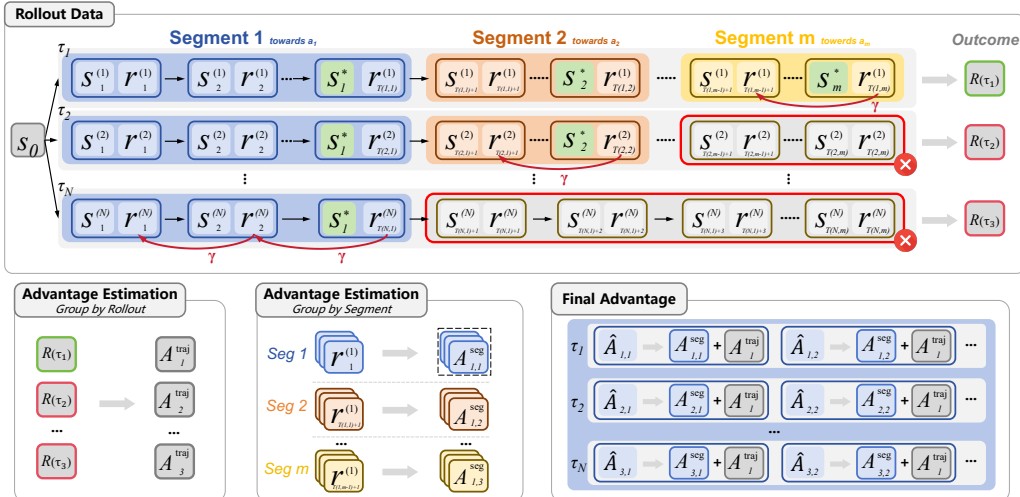

*Figure 3.* **The BEACON framework. Top:** Trajectory partitioning divides rollouts into segments at milestone boundaries; temporal reward decay (factor $\gamma$) assigns higher credit to actions closer to milestone completion. **Bottom:** Dual-scale advantage estimation computes trajectory-level advantages by comparing terminal outcomes (left), segment-level advantages by comparing returns within milestone-matched groups (middle), and combines both scales for final credit assignment (right).

### 3.4. Dual-Scale Advantage Estimation

Temporal reward shaping provides dense signal but does not fully resolve credit misattribution: actions in early segments may still receive credit influenced by outcomes in later segments through trajectory-level comparison. We address this through dual-scale advantage estimation.

**Trajectory-Level Advantage.** For a group of $G$ trajectories $\{\tau_i\}_{i=1}^{G}$ sampled for the same task, the trajectory-level advantage follows GRPO:

$$A_i^{\text{traj}} = \frac{R(\tau_i) - \mu}{\sigma + \epsilon}, \tag{4}$$

where $\mu$ and $\sigma$ are the mean and standard deviation of terminal rewards across the group.

**Segment-Level Advantage.** Trajectory-level comparison assigns identical credit to all actions regardless of position. To isolate local action quality from downstream variance, we compare segment performance only among trajectories that reached the same milestone. Define the comparison group for milestone $k$ as $\mathcal{G}_k = \{i : K_i \geq k\}$, where $K_i$ is the number of milestones reached by trajectory $\tau_i$. The segment return is:

$$R_k^{(i)} = \sum_{t \in \text{Seg}_k^{(i)}} r_t. \tag{5}$$

The segment-level advantage compares the per-step reward against the group's average per-step return:

$$A_{i,t}^{\text{seg}} = r_t - \frac{1}{|\mathcal{G}_k|} \sum_{j \in \mathcal{G}_k} \frac{R_k^{(j)}}{|\text{Seg}_k^{(j)}|}, \quad t \in \text{Seg}_k^{(i)}. \tag{6}$$

By comparing only among trajectories that reached milestone $k$, this advantage isolates the quality of actions within segment $k$ from variance in subsequent segments:

**Proposition 3.2** (Variance Isolation). *Under Assumption A.1, for trajectories in comparison group $\mathcal{G}_k$:*

$$\text{Cov}_{i \in \mathcal{G}_k}(A_{i,t}^{\text{seg}}, R_{k'}^{(i)}) \approx 0, \quad \forall i \in \mathcal{G}_k, \forall t \in \text{Seg}_k^{(i)}, \forall k' > k. \tag{7}$$

The proof is provided in Appendix A.1. This result ensures that credit for actions in segment $k$ is not corrupted by variance in later segments, directly addressing credit misattribution.

**Combined Advantage.** The final advantage for action $a_t$ in segment $\text{Seg}_k$ of trajectory $\tau_i$ is:

$$\hat{A}_{i,t} = A_i^{\text{traj}} + \lambda \cdot A_{i,t}^{\text{seg}}, \tag{8}$$

where $\lambda > 0$ balances global task performance and local segment quality.

### 3.5. Optimization

We optimize the policy using a clipped surrogate objective:

$$\mathcal{J}(\theta) = \mathbb{E}\left[\sum_t \min\left(\rho_t \hat{A}_{i,t}, \text{clip}(\rho_t, 1-\epsilon, 1+\epsilon)\hat{A}_{i,t}\right)\right] \tag{9}$$

where $\rho_t = \pi_\theta(a_t|s_t)/\pi_{\theta_{\text{old}}}(a_t|s_t)$ is the importance ratio. The complete procedure is summarized in Algorithm 1.

*Table 1.* **Main Results.** Performance comparison across benchmarks. By utilizing structural milestones, **BEACON** achieves state-of-the-art performance, showing particular robustness in Long-horizon tasks on ALFWorld.

| Type | Method | ALFWorld | | | | SciWorld | | WebShop | |
|---|---|---|---|---|---|---|---|---|---|
| | | Short | Medium | Long | Avg | Score | Succ | Score | Succ |
| *Closed-Source Models* | | | | | | | | | |
| Prompting | GPT-4o (ReAct) | 71.4 | 33.7 | 49.8 | 48.0 | 54.3 | 45.4 | 31.8 | 23.7 |
| Prompting | Gemini-2.5-Pro (ReAct) | 84.8 | 50.7 | 58.7 | 60.3 | 47.8 | 36.7 | 42.5 | 35.9 |
| *Base: Qwen2.5-1.5B-Instruct* | | | | | | | | | |
| Prompting | Direct Prompt | 5.8 | 5.1 | 0.0 | 4.1 | 5.9 | 0.7 | 23.1 | 5.2 |
| Prompting | ReAct (Yao et al., 2023) | 18.2 | 10.5 | 2.0 | 12.8 | 9.0 | 1.2 | 40.1 | 11.3 |
| Prompting | Reflexion (Shinn et al., 2023) | 31.8 | 18.9 | 3.7 | 21.8 | 7.1 | 3.9 | 55.8 | 21.9 |
| RL Training | PPO (Schulman et al., 2017) | 58.2 | 54.0 | 47.4 | 54.4 | 29.3 | 10.9 | 73.8 | 51.5 |
| RL Training | RLOO (Ahmadian et al., 2024) | 78.7 | 67.4 | 56.9 | 69.7 | - | - | 73.9 | 52.1 |
| RL Training | GRPO (Shao et al., 2024) | 76.7 | 73.9 | 53.5 | 72.8 | 31.7 | 21.1 | 75.8 | 56.8 |
| RL Training | GiGPO (Feng et al., 2025) | 90.7 | 84.3 | 79.5 | 86.1 | 35.6 | 25.8 | 83.1 | 65.0 |
| RL Training | **BEACON (Ours)** | **96.8**$_{+6.1}$ | **87.0**$_{+2.7}$ | **92.9**$_{+13.4}$ | **91.4**$_{+5.3}$ | **58.9**$_{+23.3}$ | **45.3**$_{+19.5}$ | **86.1**$_{+3.0}$ | **75.6**$_{+10.6}$ |
| *Base: Qwen2.5-7B-Instruct* | | | | | | | | | |
| Prompting | Direct Prompt | 30.2 | 10.3 | 3.2 | 14.8 | 11.4 | 4.2 | 26.4 | 7.8 |
| Prompting | ReAct (Yao et al., 2023) | 45.0 | 23.4 | 17.6 | 31.2 | 17.4 | 7.8 | 46.2 | 19.5 |
| Prompting | Reflexion (Shinn et al., 2023) | 56.5 | 38.4 | 23.8 | 42.7 | 23.4 | 11.7 | 58.1 | 28.8 |
| RL Training | PPO (Schulman et al., 2017) | 84.6 | 87.3 | 68.8 | 80.4 | 37.1 | 24.0 | 81.4 | 68.7 |
| RL Training | RLOO (Ahmadian et al., 2024) | 85.1 | 80.2 | 48.9 | 75.5 | - | - | 80.3 | 65.7 |
| RL Training | GRPO (Shao et al., 2024) | 84.1 | 79.7 | 64.7 | 77.6 | 61.8 | 49.1 | 79.3 | 66.1 |
| RL Training | GiGPO (Feng et al., 2025) | 93.6 | 91.8 | 79.2 | 90.8 | 69.2 | 53.4 | 84.4 | 72.8 |
| RL Training | **BEACON (Ours)** | **95.1**$_{+1.5}$ | **94.9**$_{+3.1}$ | **90.0**$_{+10.8}$ | **94.5**$_{+3.7}$ | **83.7**$_{+14.5}$ | **64.3**$_{+10.9}$ | **87.7**$_{+3.3}$ | **79.7**$_{+6.9}$ |

## 4. Experiments

### 4.1. Experimental Setup

**Benchmarks.** We evaluate on three long-horizon benchmarks: ALFWorld (Shridhar et al., 2021), Science-World (Wang et al., 2022) and WebShop (Yao et al., 2022). *ALFWorld* is a text-based embodied environment where agents complete household tasks (e.g., heating objects, cleaning items) through multi-step interaction, receiving only sparse terminal rewards upon task completion. *Web-Shop* is a web navigation environment with 1.18M products, requiring agents to search, filter, and purchase items matching natural language specifications through compositional understanding of product attributes. *ScienceWorld* is a text-based environment for scientific reasoning, spanning 30 task types across 10 domains, requiring agents to conduct virtual experiments (e.g., measuring melting points, testing electrical conductivity). See Appendix C.1 for details.

**Baselines.** We compare against baselines across paradigms: (1) Closed-source models: GPT-4o (Hurst et al., 2024) and Gemini-2.5-Pro (Comanici et al., 2025), evaluated under ReAct (Yao et al., 2023) prompting as reference points for frontier model capabilities. (2) Prompting methods: ReAct, which guides multi-step reasoning through in-context chain-of-thought without training. (3) RL training methods: PPO (Schulman et al., 2017), a standard actor-critic algorithm, and group-based

approaches GRPO (Shao et al., 2024) and GiGPO (Feng et al., 2025), which estimate advantages over trajectory groups without learned critics.

**Implementation.** We use Qwen2.5-1.5B-Instruct and Qwen2.5-7B-Instruct (Qwen et al., 2025) as base models. For fair comparison, all RL methods use identical training configurations. BEACON-specific parameters ($\gamma = 0.95$, $\lambda = 1.0$) are fixed across all benchmarks without task-specific tuning. Full details are in Appendix C.4.

### 4.2. Main Results

**Overall Performance.** Table 1 presents results. BEA-CON achieves the highest success rate across all benchmarks and model scales. On ALFWorld with the 1.5B model, BEACON achieves 91.4% average success rate, surpassing GiGPO (86.1%) by 5.3% and GRPO (72.8%) by 18.6%. On WebShop, BEACON achieves 75.6% success rate compared to 65.0% for GiGPO and 56.8% for GRPO. On ScienceWorld, BEACON reaches 45.3% success versus 25.8% for GiGPO and 21.1% for GRPO. Scaling to Qwen2.5-7B yields consistent improvements: BEACON achieves 94.5% on ALFWorld and 79.7% on WebShop. Notably, even the 1.5B BEACON model outperforms closed-source baselines (GPT-4o: 48.0% on ALFWorld, 23.7% on WebShop), demonstrating that milestone-anchored credit assignment provides advantages that model scale alone cannot match. We provide task-wise breakdown for ALFWorld in

**Algorithm 1** BEACON Training

**Require:** Policy $\pi_\theta$, milestone detector $\Phi$, group size $G$, decay $\gamma$, weight $\lambda$

1: **for** each iteration **do**
2:     // Sample trajectories
3:     Sample $G$ trajectories $\{\tau_i\}_{i=1}^G$ using $\pi_\theta$
4:     **for** each trajectory $\tau_i$ **do**
5:         // Detect milestones and partition
6:         $\mathcal{M}_i \leftarrow \{t : \Phi(s_t, a_t, s_{t+1}) = 1\}$
7:         Partition $\tau_i$ into $\{\text{Seg}_k^{(i)}\}_{k=1}^{K_i+1}$ using $\mathcal{M}_i$
8:         // Compute shaped rewards
9:         $r_t \leftarrow \mathbb{I}[k \le K_i] \cdot R_{\text{ms}} \cdot \gamma^{t_k - t}$ for each $t \in \text{Seg}_k^{(i)}$
10:    **end for**
11:    // Compute trajectory-level advantages
12:    $\mu \leftarrow \frac{1}{G}\sum_i R(\tau_i), \quad \sigma \leftarrow \text{std}(\{R(\tau_i)\})$
13:    $A_i^{\text{traj}} \leftarrow (R(\tau_i) - \mu)/(\sigma + \epsilon)$ for all $i$
14:    // Compute segment-level advantages
15:    **for** $k = 1, \ldots, \max_i K_i$ **do**
16:        $\mathcal{G}_k \leftarrow \{i : K_i \ge k\}$
17:        $A_{i,t}^{\text{seg}} \leftarrow r_t - \frac{1}{|\mathcal{G}_k|}\sum_{j \in \mathcal{G}_k} R_k^{(j)}/|\text{Seg}_k^{(j)}|$ for $t \in \text{Seg}_k^{(i)}, i \in \mathcal{G}_k$
18:    **end for**
19:    // Combine advantages and update policy
20:    $\hat{A}_{i,t} \leftarrow A_i^{\text{traj}} + \lambda \cdot A_{i,t}^{\text{seg}}$ for each $a_t \in \text{Seg}_k^{(i)}$
21:    Update $\theta$ by maximizing $\mathcal{J}(\theta)$
22: **end for**

---

Appendix B, showing consistent gains across all task types.

**Horizon-Dependent Performance.** On ALFWorld with the 1.5B model, GRPO exhibits severe degradation as horizon extends: success rate drops from 76.7% on Short tasks to 53.5% on Long tasks, a 30% relative decline. GiGPO mitigates this partially (90.7% to 79.5%, 12.4% relative decline) but still shows clear degradation. In contrast, BEACON maintains robust performance across horizons (96.8% Short, 87.0% Medium, 92.9% Long). Figure 5(b) illustrates this pattern on the 7B model through relative improvement over GRPO. On Short tasks, BEACON and GiGPO achieve comparable gains (+13% vs +11%). However, the gap widens as horizon extends: on Long tasks, BEACON reaches +39% while GiGPO remains at +22%. GiGPO relies on state recurrence for step-level grouping, which diminishes as policies improve and trajectories diversify. These results indicate that milestone-anchored credit assignment provides increasing benefit as task horizons extend.

### 4.3. Analysis

**Partial Successes Become Learning Signal.** We analyze sample efficiency by categorizing trajectories during training into three types: full successes (complete the task), partial successes (complete at least one milestone but fail the final task), and complete failures (achieve no milestone). Figure 4 shows the distribution on ALFWorld (Qwen2.5-1.5B) across 150 training iterations. Under GRPO, 39% of

trajectories at iteration 150 are partial successes that complete at least one milestone but receive zero reward. GiGPO reduces this to 28% through state-based grouping, but substantial signal remains discarded. BEACON's temporal reward shaping provides positive reward for milestone completion, reducing partial successes to 13%. Effective sample utilization improves from 23.7% to 82.0%, a $3.5\times$ increase in trajectories providing useful gradient signal.

**Gradient Starvation.** We measure the Zero-Advantage Ratio (ZAR), defined as the fraction of samples receiving near-zero advantage during training. Figure 5(a) shows ZAR on ALFWorld. GRPO starts near 100% ZAR and decreases to around 55% by iteration 150, indicating that over half of samples provide no learning signal even after extended training. BEACON starts at 45% ZAR and rapidly decreases to approximately 10%, confirming that milestone-anchored credit assignment substantially alleviates gradient starvation by extracting signal from partial successes.

**Credit Concentration.** We compute the Credit Concentration Ratio (CCR), defined as the average advantage magnitude for milestone actions divided by that for non-milestone actions. CCR=1 indicates uniform credit; CCR>1 indicates concentration on milestones. Figure 6(a) shows CCR across methods on ALFWorld (Qwen2.5-1.5B). GiGPO exhibits the highest CCR (2.36), meaning milestone actions receive $2.36\times$ more credit than non-milestone actions. GRPO shows moderate concentration (1.37). BEACON has the lowest CCR (0.84), indicating that non-milestone actions receive slightly more credit than milestone actions. Despite lower concentration, BEACON achieves the highest performance. This suggests that credit concentration penalizes intermediate actions necessary for reaching milestones. BEACON's temporal decay assigns graduated positive credit to all actions within successful segments, preserving signal for exploratory steps that enable milestone completion.

**Beyond Behavior Cloning.** A potential concern is whether BEACON degrades to behavior cloning given its use of milestone structure. Figure 6(b) compares BEACON against supervised fine-tuning (SFT) on oracle trajectories on ALFWorld (Qwen2.5-1.5B). Supervised fine-tuning on oracle trajectories achieves 43% success rate. BEACON with $\gamma=0$ (milestone reward only) reaches 81%, demonstrating that milestone-anchored credit assignment alone enables the policy to discover strategies superior to the oracle. Introducing temporal decay ($\gamma=0.95$) further improves performance to 91.4%. This confirms that the milestone structure provides credit assignment anchors, but the policy discovers execution strategies superior to the oracle trajectories.

**Training Dynamics.** Figure 7 compares training dynamics on ALFWorld (Qwen2.5-1.5B). BEACON converges

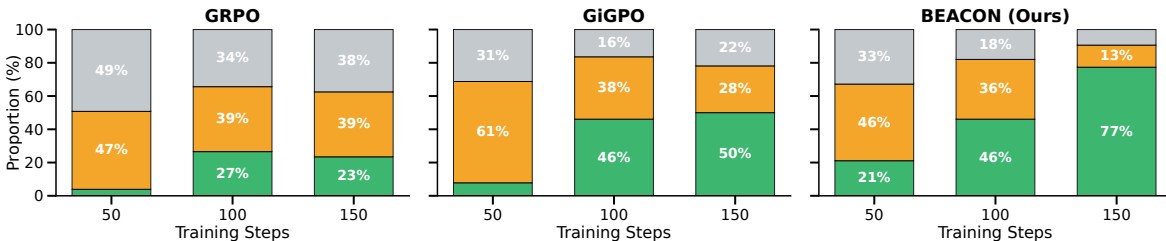

*Figure 4.* **Sample Efficiency.** Trajectory distribution during training on ALFWorld. Green: full successes; Orange: partial successes (complete $\geq 1$ milestone but fail); Gray: complete failures.

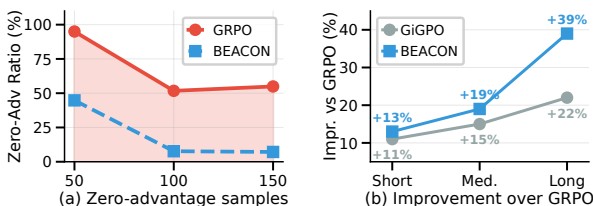

*Figure 5.* **Learning Signal and Horizon Scaling.** (a) Zero-Advantage Ratio during training. (b) Relative improvement over GRPO by task horizon.

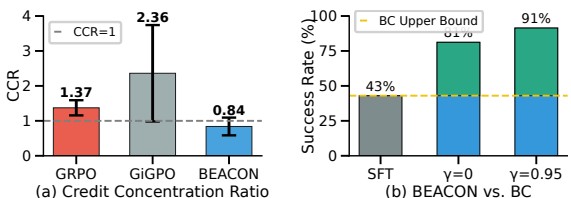

*Figure 6.* **Credit Distribution and Policy Optimization.** (a) Credit Concentration Ratio across methods. Higher CCR indicates more aggressive concentration on milestone actions. (b) Comparison with behavior cloning (SFT on oracle trajectories).

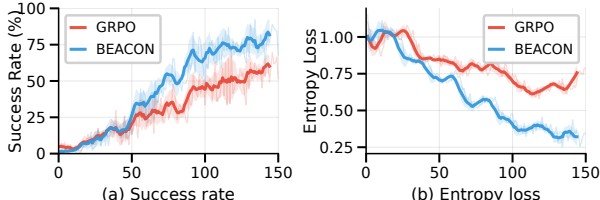

*Figure 7.* **Training Dynamics.** (a) Success rate. BEACON converges faster than GRPO. (b) Policy entropy evolution. BEACON exhibits smooth reduction indicating stable refinement.

*Table 2.* **Ablation Study with Qwen2.5-1.5B-Instruct.**

| Variant | ALFWorld | WebShop |
|---|---|---|
| **BEACON** | **91.4** | **75.6** |
| *Trajectory Partitioning* | | |
| w/ 50% Milestone Dropout | 82.8 | 68.3 |
| w/ Random Partition | 74.2 | 57.6 |
| *Reward Shaping* | | |
| w/o Temporal Decay ($\gamma = 0$) | 81.2 | 62.1 |
| w/ Uniform Shaping ($\gamma = 1$) | 71.8 | 64.1 |
| *Advantage Estimation* | | |
| w/o Segment-level Adv | 72.8 | 56.8 |
| w/o Trajectory-level Adv | 23.4 | 67.9 |
| GRPO | 72.8 | 56.8 |

faster: it reaches 60% success rate by iteration 50, while GRPO requires iteration 120 to reach the same threshold. This faster convergence is consistent with BEACON's improved sample utilization (23.7% to 82.0%), as more trajectories contribute useful gradient signal per batch. Figure 7(b) shows policy entropy. BEACON exhibits smooth entropy reduction, while GRPO maintains high entropy throughout. The contrast reflects the difference in gradient quality: BEACON receives consistent feedback from milestone completion, enabling steady policy refinement.

### 4.4. Ablation Study

**Trajectory Partitioning.** We evaluate degraded partitioning strategies on ALFWorld. Random partitioning (selecting 5 arbitrary positions as milestones) achieves 74.2%, slightly above GRPO (72.8%), indicating that segmentation structure itself provides modest benefit. With 50% milestone dropout, performance degrades gracefully to 82.8%, still outperforming GRPO by 10%, indicating that BEACON tolerates imperfect milestone detection. Notably, the gap

between random and full milestones (17.2%) far exceeds the gap between GRPO and random (1.4%), demonstrating that BEACON's gains stem primarily from exploiting task-inherent structure rather than segmentation alone.

**Temporal Reward Shaping.** Removing temporal decay ($\gamma = 0$) reduces performance from 91.4% to 81.2% on ALFWorld and from 75.6% to 62.1% on WebShop, yet still outperforms GRPO (72.8% and 56.8%). This confirms that milestone-anchored structure itself provides significant benefit, while temporal decay contributes additional gains by distinguishing action contributions within segments. Notably, uniform shaping ($\gamma = 1$) performs worse than no shaping on ALFWorld (71.8% vs 81.2%): assigning equal credit to all actions obscures the distinction between critical and preparatory actions, producing misleading gradients.

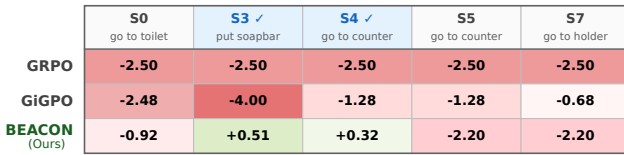
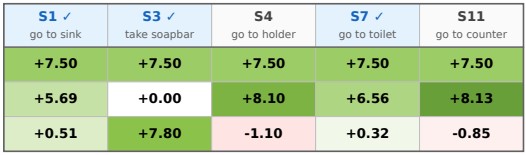

|  | S0
go to toilet | S3 ✓
put soapbar | S4 ✓
go to counter | S5
go to counter | S7
go to holder |
|---|---|---|---|---|---|
| **GRPO** | -2.50 | -2.50 | -2.50 | -2.50 | -2.50 |
| **GiGPO** | -2.48 | -4.00 | -1.28 | -1.28 | -0.68 |
| **BEACON**
(Ours) | -0.92 | +0.51 | +0.32 | -2.20 | -2.20 |

(a) Failed Trajectory

|  | S1 ✓
go to sink | S3 ✓
take soapbar | S4
go to holder | S7 ✓
go to toilet | S11
go to counter |
|---|---|---|---|---|---|
| **GRPO** | +7.50 | +7.50 | +7.50 | +7.50 | +7.50 |
| **GiGPO** | +5.69 | +0.00 | +8.10 | +6.56 | +8.13 |
| **BEACON**
(Ours) | +0.51 | +7.80 | -1.10 | +0.32 | -0.85 |

(b) Successful Trajectory

*Figure 8.* **Credit Assignment on Representative Trajectories.** (a) Failed trajectory with intermediate milestones. (b) Successful trajectory with detours. GRPO assigns uniform credit to all actions; GiGPO produces counterintuitive assignments due to state-based grouping; BEACON credits milestone completions while appropriately penalizing errors and inefficient detours.

**Dual-Scale Advantage.** Removing segment-level advantage naturally degrades BEACON to GRPO (72.8% on ALFWorld, 56.8% on WebShop), establishing GRPO as the performance lower bound. Removing trajectory-level advantage produces different effects across benchmarks: severe degradation on ALFWorld (23.4%) but reasonable performance on WebShop (67.9%). This difference reflects task structure. On ALFWorld, segment-level optimization alone can reinforce actions that achieve intermediate milestones but lead to eventual task failure. Trajectory-level feedback provides necessary correction. On WebShop, milestone completion aligns more directly with task success, with segment-level feedback driving the primary improvement while trajectory-level feedback provides additional gains. The dual-scale formulation leverages both signals, achieving robust performance across diverse task structures.

### 4.5. Case Study

Figure 8 presents credit assignment on two representative trajectories from ALFWorld. In the failed trajectory, the agent completes milestones S3 and S4 before failing. GRPO assigns uniform negative advantage ($A=-2.50$) to all actions. GiGPO produces counterintuitive credit: milestone S3 receives the lowest advantage ($A=-4.00$), because state-based grouping compares it against successful trajectories. BEACON credits milestones ($A=+0.51$) while penalizing errors. In the successful trajectory with an unnecessary detour at S4, GRPO assigns uniform positive advantage ($A=+7.50$). GiGPO rewards the detour most heavily ($A=+8.10$). BEACON penalizes the detour ($A=-1.10$) while crediting milestones. These examples illustrate how BEACON provides precise credit assignment that distinguishes productive actions from errors and inefficiencies.

## 5. Related Work

Our work relates to policy optimization for language models and credit assignment in reinforcement learning.

**Policy Optimization for Language Models.** PPO (Schulman et al., 2017; Ouyang et al., 2022) is widely used for RLHF but requires a value network that struggles over long horizons. Critic-free methods such as DPO (Rafailov et al.,

2023), GRPO (Shao et al., 2024), and RLOO (Ahmadian et al., 2024) eliminate this overhead and achieve strong results on reasoning tasks (Guo et al., 2025; Yu et al., 2026; Li et al., 2025). However, when applied to LLM agents for web navigation (Deng et al., 2023; Zhou et al., 2024a; Qi et al., 2025), embodied control (Shridhar et al., 2021; Wang et al., 2022; Li et al., 2026), and tool use (Schick et al., 2023; Qin et al., 2024; Wang et al., 2025; Zeng et al., 2024; Chen et al., 2023), these trajectory-level methods assign identical credit to all actions regardless of individual contribution, causing performance degradation as task horizons extend. BEACON exploits semantic milestones inherent to agentic tasks, enabling segment-level comparison within trajectories.

**Credit Assignment and Reward Shaping.** Existing approaches to finer-grained credit assignment introduce distinct limitations. Auxiliary model methods, including process reward models (Lightman et al., 2024; Wang et al., 2024), utterance-level critics (Zhou et al., 2024b), implicit reward models (Cui et al., 2025), and co-evolving verifiers (Pan et al., 2026), require expensive annotation, risk reward hacking (Gao et al., 2022), or add training complexity. Monte Carlo methods (Kazemnejad et al., 2025) avoid learned models but incur substantial sampling overhead from multiple rollouts per step. Structure-based methods such as GiGPO (Feng et al., 2025) and RLVMR (Zhang et al., 2026b) exploit repeated states or reasoning patterns for localized comparison, but depend on incidental structure that may be sparse in long-horizon tasks. BEACON instead anchors credit to milestones that directly reflect task progress, providing consistent segment-level comparison without auxiliary models, sampling overhead, or reliance on emergent trajectory patterns.

## 6. Conclusion

We introduced BEACON, a framework that addresses credit misattribution and sample inefficiency in trajectory-level policy optimization for long-horizon language agents. BEACON exploits the compositional structure of long-horizon tasks: milestones, observable state transitions indicating subgoal completion, exhibit an approximate Markov property that enables credit to be decoupled across segments. By partitioning trajectories at milestone boundaries, ap-

plying temporal reward shaping within segments, and estimating advantages at dual scales, BEACON isolates local action quality from downstream variance. Experiments on ALFWorld, WebShop, and ScienceWorld demonstrate improvements that amplify as task horizons extend, with effective sample utilization improvement. These results establish milestone-anchored credit assignment as an effective paradigm for training long-horizon language agents. We further discuss the limitations of BEACON and its future directions in Appendix D.

## Impact Statement

This paper presents work whose goal is to advance the training of language model agents for long-horizon tasks. The primary societal impact is enabling more capable autonomous agents that can assist humans in complex, multi-step tasks such as web navigation, household management, and scientific experimentation. While improved agent capabilities could increase productivity and accessibility, they also raise considerations around automation of tasks currently performed by humans. Our method does not introduce new capabilities beyond existing language models but rather improves the efficiency of training agents on tasks with sparse rewards. We do not anticipate specific negative societal consequences beyond those generally associated with advances in language model agents.

## Acknowledgements

This work was supported by New Generation Artificial Intelligence-NationalScience and Technology Major Project (2025ZD0123102), National Natural Science Foundation of China (No. 62506332), "Pioneer" and "Leading Goose" R&D Program of Zhejiang (NO. 2026C02A1223), and CCF-Baidu Open Fund.

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

# A. Theoretical Analysis

This appendix provides formal analysis supporting the design of BEACON, establishing that segment-level advantages isolate local action quality from downstream variance.

## A.1. Variance Isolation in Segment-Level Advantages

The foundation of BEACON's credit assignment is the structural assumption that milestone states are approximately Markovian.

**Assumption A.1** (Milestone Markov Property). For milestone state $s_{t_k}$ reached at timestep $t_k$:

$$P(\text{Seg}_{k+1}, \ldots, \text{Seg}_{K+1} \mid s_{t_k}, \text{Seg}_1, \ldots, \text{Seg}_k)$$
$$\approx P(\text{Seg}_{k+1}, \ldots, \text{Seg}_{K+1} \mid s_{t_k}). \quad (10)$$

This assumption is natural for compositional tasks: once a subgoal is achieved (e.g., an object is picked up), subsequent success depends on completing remaining subgoals, not on how previous subgoals were achieved.

**Proposition A.2** (Variance Isolation). *Under Assumption A.1, for trajectories in comparison group $\mathcal{G}_k = \{i : K_i \geq k\}$:*

$$\text{Cov}_{i \in \mathcal{G}_k}(A_{i,t}^{\text{seg}}, R_{k'}^{(i)}) \approx 0, \quad \forall i \in \mathcal{G}_k, \forall t \in \text{Seg}_k^{(i)}, \forall k' > k. \quad (11)$$

*Proof.* For trajectories in $\mathcal{G}_k$, the per-step segment-level advantage is:

$$A_{i,t}^{\text{seg}} = r_t - \bar{b}_k, \quad \text{where } \bar{b}_k = \frac{1}{|\mathcal{G}_k|} \sum_{j \in \mathcal{G}_k} \frac{R_k^{(j)}}{|\text{Seg}_k^{(j)}|}. \quad (12)$$

For $t \in \text{Seg}_k^{(i)}$, the shaped reward $r_t$ depends only on the position within segment $k$ (through $t_k^{(i)} - t$) and on actions $\{a_{t'} : t' \in \text{Seg}_k^{(i)}\}$, which occur before milestone $k$ is reached. For $k' > k$, the segment return $R_{k'}^{(i)}$ depends only on actions $\{a_t : t \in \text{Seg}_{k'}^{(i)}\}$, which occur after milestone $k$ is reached.

By Assumption A.1, conditioned on the milestone state $s_{t_k}$, the actions in segment $k'$ are independent of the actions in segment $k$:

$$\mathbb{E}[r_t \cdot R_{k'}^{(i)} \mid i \in \mathcal{G}_k] \approx \mathbb{E}[r_t \mid i \in \mathcal{G}_k] \cdot \mathbb{E}[R_{k'}^{(i)} \mid i \in \mathcal{G}_k]. \quad (13)$$

Since $\bar{R}_k$ is constant over $\mathcal{G}_k$:

$$\text{Cov}(A_{i,k}^{\text{seg}}, R_{k'}^{(i)}) = \text{Cov}(R_k^{(i)} - \bar{R}_k, R_{k'}^{(i)})$$
$$= \text{Cov}(R_k^{(i)}, R_{k'}^{(i)}) \approx 0. \quad (14)$$

$\square$

This result establishes that segment-level advantages isolate local action quality from downstream variance: the gradient for actions in segment $k$ is not affected by outcomes in later segments, directly addressing credit misattribution.

## A.2. Discussion of Assumptions

The Milestone Markov Property (Assumption A.1) is central to the variance isolation guarantee. This assumption holds well when milestone states encode complete subgoal achievement and future success depends primarily on remaining subgoals rather than execution details of past subgoals.

The assumption may be approximate when resources carry across segments (e.g., inventory limits) or when execution efficiency affects future success (e.g., time constraints). However, even when the Markov property is only approximately satisfied, BEACON provides empirical benefits: partial successes still contribute gradient signal through shaped rewards, and segment-level comparison reduces downstream variance even if it does not fully eliminate it. The trajectory-level advantage component maintains task alignment regardless of the Markov property. The experimental results in Section 4 demonstrate substantial improvements on tasks where the assumption is only approximately satisfied.

# B. Task-wise Analysis on ALFWorld

We report the success rates of different methods across all six ALFWorld task types in Table 3. The table presents results for both Qwen2.5-1.5B and Qwen2.5-7B base models. BEACON consistently outperforms other methods on both model scales, with particularly strong gains on Pick2 (+13% on 1.5B, +11% on 7B), which requires locating and picking up two separate objects and thus involves more milestones for credit assignment. Notably, BEACON-trained 1.5B models (91.4%) substantially outperform GPT-4o (48.0%) and Gemini-2.5-Pro (60.3%), demonstrating that task-specific training with proper credit assignment can surpass general-purpose large models.

# C. Experimental Details

## C.1. Benchmark Descriptions

We evaluate BEACON on three diverse benchmarks spanning embodied reasoning, web navigation, and scientific experimentation.

**ALFWorld.** ALFWorld (Shridhar et al., 2021) is a text-based embodied reasoning benchmark that aligns TextWorld (Côté et al., 2018) environments with AL-FRED (Shridhar et al., 2020) visual tasks. The benchmark comprises six task types: PICK (pick up an object),

*Table 3.* **ALFWorld Task-wise Results.** Success rate (%) on each task type.

| Type | Method | ALFWorld | | | | | | |
|---|---|---|---|---|---|---|---|---|
| | | Pick | Look | Clean | Heat | Cool | Pick2 | All |
| *Closed-Source Models* | | | | | | | | |
| Prompting | GPT-4o | 75.3 | 60.8 | 31.2 | 56.7 | 21.6 | 49.8 | 48.0 |
| Prompting | Gemini-2.5-Pro | 92.8 | 63.3 | 62.1 | 69.0 | 26.6 | 58.7 | 60.3 |
| *Base: Qwen2.5-1.5B-Instruct* | | | | | | | | |
| Prompting | Direct Prompt | 5.9 | 5.5 | 3.3 | 9.7 | 4.2 | 0.0 | 4.1 |
| Prompting | ReAct | 17.4 | 20.5 | 15.7 | 6.2 | 7.7 | 2.0 | 12.8 |
| Prompting | Reflexion | 35.3 | 22.2 | 21.7 | 13.6 | 19.4 | 3.7 | 21.8 |
| RL Training | PPO | 64.8 | 40.5 | 57.1 | 60.6 | 46.4 | 47.4 | 54.4 |
| RL Training | RLOO | 88.3 | 52.8 | 71.0 | 62.8 | 66.4 | 56.9 | 69.7 |
| RL Training | GRPO | 85.3 | 53.7 | 84.5 | 78.2 | 59.7 | 53.5 | 72.8 |
| RL Training | GiGPO | 96.0 | 76.5 | **91.8** | 91.3 | 71.7 | 79.5 | 86.1 |
| RL Training | **BEACON (Ours)** | **100** | **88.2** | 86.7 | **100** | **78.9** | **92.9** | **91.4** |
| | $\Delta$ *vs GRPO* | +14.7 | +34.5 | +2.2 | +21.8 | +19.2 | +39.4 | +18.6 |
| *Base: Qwen2.5-7B-Instruct* | | | | | | | | |
| Prompting | Direct Prompt | 33.4 | 21.6 | 19.3 | 6.9 | 2.8 | 3.2 | 14.8 |
| Prompting | ReAct | 48.5 | 35.4 | 34.3 | 13.2 | 18.2 | 17.6 | 31.2 |
| Prompting | Reflexion | 62.0 | 41.6 | 44.9 | 30.9 | 36.3 | 23.8 | 42.7 |
| RL Training | PPO | 92.3 | 64.0 | 92.5 | 89.5 | 80.3 | 68.8 | 80.4 |
| RL Training | RLOO | 87.6 | 78.2 | 87.3 | 81.3 | 71.9 | 48.9 | 75.5 |
| RL Training | GRPO | 90.8 | 66.1 | 89.3 | 74.7 | 72.5 | 64.7 | 77.6 |
| RL Training | GiGPO | 97.7 | **82.7** | **98.8** | 83.7 | 89.3 | 79.2 | 90.8 |
| RL Training | **BEACON (Ours)** | **100** | 81.8 | 96.3 | **92.9** | **94.7** | **90.0** | **94.5** |
| | $\Delta$ *vs GRPO* | +9.2 | +15.7 | +7.0 | +18.2 | +22.2 | +25.3 | +16.9 |

CLEAN (clean an object), HEAT (heat an object), COOL (cool an object), LOOK (examine an object under light), and PICK2 (pick up two objects). Tasks require agents to navigate household environments and manipulate objects through natural language commands. We use the standard train/validation/test split with 3,321/140/140 tasks respectively. Following prior work, we stratify tasks by optimal trajectory length: Short ($L^* \leq 4$), Medium ($5 \leq L^* \leq 7$), and Long ($L^* > 7$).

**WebShop.** WebShop (Yao et al., 2022) is a simulated e-commerce environment containing 1.18 million real-world products and 12,087 human instructions. Agents must navigate web pages through search, filtering, and clicking actions to purchase products matching natural language specifications. The benchmark tests compositional understanding of product attributes including color, size, price constraints, and feature requirements. We use the standard evaluation protocol with 500 test instructions and report both Score (partial credit based on attribute matching) and Success Rate (binary task completion).

**ScienceWorld.** ScienceWorld (Wang et al., 2022) presents 30 scientific reasoning tasks requiring agents to conduct virtual experiments, such as measuring melting points, testing electrical conductivity, and identifying life stages of organisms. Tasks involve long action sequences frequently exceeding 30 steps, with complex dependencies between

sub-experiments. The environment provides explicit sub-goal feedback that our milestone detector directly consumes. We report both Score (normalized progress) and Success Rate across all 30 task types.

### C.2. Diagnostic Metrics

We introduce two metrics to quantify credit assignment quality in policy optimization.

**Contradictory Action Ratio (CAR).** For a batch of trajectories, let $\mathcal{S}_{\text{shared}}$ denote the set of state-action pairs $(s, a)$ that appear in multiple trajectories. For each $(s, a) \in \mathcal{S}_{\text{shared}}$, let $A^+$ and $A^-$ denote the number of trajectories where this pair receives positive and negative advantages, respectively. The CAR is defined as:

$$\text{CAR} = \frac{1}{|\mathcal{S}_{\text{shared}}|} \sum_{(s,a) \in \mathcal{S}_{\text{shared}}} \mathbb{I}[A^+ > 0 \wedge A^- > 0], \quad (15)$$

where $\mathbb{I}[\cdot]$ is the indicator function. CAR measures the fraction of repeated state-action pairs receiving contradictory gradient signals.

**Effective Gradient Ratio (EGR).** For each state-action pair $(s, a) \in \mathcal{S}_{\text{shared}}$, let $g^+$ and $g^-$ denote the sum of positive and negative advantage magnitudes, respectively. The

EGR is defined as:

$$\text{EGR} = \frac{\sum_{(s,a)\in\mathcal{S}_{\text{shared}}} |g^+ - g^-|}{\sum_{(s,a)\in\mathcal{S}_{\text{shared}}} (g^+ + g^-)}. \tag{16}$$

EGR measures the proportion of gradient magnitude that survives after cancellation from contradictory signals. An EGR of 1.0 indicates fully consistent gradients, while lower values indicate greater cancellation.

## C.3. Implementation Details

All experiments are conducted using the veRL framework (Sheng et al., 2025) with vLLM (Kwon et al., 2023) for efficient inference. We use 8 NVIDIA A100 80GB GPUs for training. Gradient checkpointing is enabled to reduce memory consumption. The reference model uses CPU parameter offloading while the actor model remains fully on GPU. Training 150 iterations takes approximately 10 hours for ALFWorld and ScienceWorld, and 8 hours for WebShop.

For all group-based methods (GRPO, GiGPO, BEACON), we use identical base configurations to ensure fair comparison. The only differences are in the advantage computation mechanisms specific to each method. All experiments use a fixed random seed (seed=0). Evaluation is conducted on 128 samples per checkpoint.

## C.4. Hyperparameters

Table 4 presents the hyperparameters used in our experiments. BEACON-specific parameters are listed separately from general training parameters shared across all methods.

## D. Limitations and Future Work

**Milestone Detection.** BEACON relies on a task-specific milestone detector $\Phi$ that identifies subgoal completions from environment feedback. In our experiments, milestones are extracted through pattern matching on environment responses (ALFWorld), page transitions (WebShop), or explicit subgoal signals (ScienceWorld). This approach requires domain knowledge to design appropriate detectors and may not generalize to environments without clear subgoal structure or verifiable state transitions. Developing automated milestone discovery methods, potentially through learning or leveraging large language models to identify semantically meaningful progress, remains an important open problem.

**Milestone Granularity.** The effectiveness of BEACON depends on milestones occurring at an appropriate granularity. If milestones are too sparse, BEACON approaches trajectory-level optimization; if too dense, the segment-level advantages may become noisy. Our experiments use natu-

*Table 4.* **Hyperparameters.** BEACON-specific parameters control milestone-anchored credit assignment; other parameters are shared across all group-based methods (GRPO, GiGPO, BEACON) for fair comparison. For environment-specific values, we report ALFWorld / WebShop / ScienceWorld.

| Hyperparameter | Symbol | Value |
|---|---|---|
| *BEACON-specific* | | |
| Segment advantage weight | $\lambda$ | 1.0 |
| Temporal decay factor | $\gamma$ | 0.95 |
| *Optimization* | | |
| Learning rate | – | $1 \times 10^{-6}$ |
| PPO clip ratio | $\epsilon$ | 0.2 |
| Gradient clip norm | – | 1.0 |
| Entropy coefficient | – | 0.001 |
| KL penalty coefficient | $\beta$ | 0.01 |
| *Batch Configuration* | | |
| Prompts per iteration | – | 16 |
| Rollouts per prompt | $G$ | 8 |
| PPO mini-batch size | – | 256 |
| *Sequence* | | |
| Max prompt length | – | 7000 |
| Max response length | – | 512 |
| Temperature (train / eval) | – | 1.0 / 0.4 |
| *Environment* | | |
| Max steps per episode | $T$ | 30 / 15 / 30 |
| Total training iterations | – | 150 |

rally occurring task milestones without tuning granularity, but optimal milestone density likely varies across tasks. Investigating adaptive or hierarchical milestone structures could further improve performance.

**Benchmark Scope.** We evaluate BEACON on three benchmarks spanning embodied reasoning, web navigation, and scientific experimentation. While these cover diverse agent capabilities, all involve discrete action spaces and text-based interaction. The applicability of milestone-anchored credit assignment to continuous control, multi-agent settings, or tasks with less compositional structure remains unexplored.

