# OpenReview forum: "Milestone-Guided Policy Learning for Long-Horizon Language Agents"
_ICML.cc/2026/Conference — ICML 2026 regular_

### Official Review · Reviewer_KKWY · 2026-03-09

**Soundness:** 3
**Presentation:** 3
**Significance:** 3
**Originality:** 2
**Overall Recommendation:** 5
**Confidence:** 3

**Summary:**

This paper introduces BEACON, a milestone-guided policy learning framework designed to improve the training of large language model (LLM) agents on long-horizon tasks. The authors identify that traditional trajectory-level reinforcement learning (RL) methods, such as GRPO, suffer from systemic performance collapse as task horizons extend.

BEACON enhances long-horizon agent training by breaking down complex tasks into manageable parts:Trajectory Partitioning》》Temporal Reward Shaping》》Dual-Scale Advantage Estimation.  This idea looks smart.

Main Contributions and Results：
1. Performance Breakthroughs: BEACON consistently outperforms GRPO and GiGPO across ALFWorld, WebShop, and ScienceWorld. 2.Improved Efficiency: The framework improves effective sample utilization from 23.7% to 82.0% by extracting learning signals from partial successes.
3.Model Scaling and Robustness: BEACON-trained 1.5B models can outperform much larger closed-source models like GPT-4o and Gemini-2.5-Pro on specific agentic benchmarks

**Compliance With Llm Reviewing Policy:**

Affirmed.

**Final Justification:**

After considering both the paper and the authors’ rebuttal, I recommend acceptance. The paper is technically solid, clearly written, and addresses an important problem in long-horizon LLM agent training. Its main strengths are the strong empirical performance across multiple benchmarks, the practical improvement over GRPO-style methods, and a well-motivated framework that makes better use of partial success signals. In terms of originality, I still view the contribution more as a thoughtful integration and adaptation of existing RL ideas than a fundamentally new RL paradigm, but I believe this combination is effective and valuable for the targeted setting.

**Key Questions For Authors:**

Generalizability:The current framework relies on a task-specific milestone indicator derived from verifiable state changes or environment feedback. In unstructured or real-world domains where environment signals are less explicit than those in ALFWorld or ScienceWorld, how does the performance of BEACON degrade if milestones must be inferred by a separate model?

Sensitivity to Milestone Granularity:You mention that if milestones are too sparse, the method approaches trajectory-level optimization, and if too dense, advantages may become noisy. Did you conduct any sensitivity analysis regarding the "optimal" number of milestones per task?

Comparison with Hierarchical Reinforcement Learning (HRL):How does BEACON conceptually and empirically differ from traditional HRL approaches that use subgoals for credit assignment?

**Limitations:**

yes

**Strengths And Weaknesses:**

Strengths

The empirical section is strong. The method is evaluated on three fairly different long-horizon agent benchmarks, and the gains over GRPO and GiGPO appear consistent rather than isolated to a single environment.

I also found the theoretical analysis helpful. In particular, the variance-isolation result gives some justification for why segment-level advantages may provide cleaner credit assignment than purely trajectory-level updates.

The paper is clearly written overall. The main idea is easy to follow, and the figures help explain how milestone partitioning and reward shaping work together.

Weaknesses

A main concern is the reliance on milestone detectors. This seems reasonable in the chosen benchmarks, but it is less clear how well the method would work when milestone signals are missing, delayed, or noisy.

The overall framework is sensible, but some of its ingredients are not fundamentally new from an RL perspective. The novelty comes more from how these ideas are combined and adapted to long-horizon LLM agents.

The experimental comparison is strong with respect to GRPO-style methods, but I would have liked a clearer discussion of how this approach relates to or differs from more classical hierarchical RL methods that also leverage subgoals or temporal abstraction

---

> ### Author Rebuttal · Authors · 2026-03-31
>
> We thank you for your time in reviewing our paper and the thoughtful questions. We respond to each point below.
>
> > Milestone Detection Robustness and Generalizability (W1, Q1)
>
> To simulate environments where milestone signals are less explicit, we removed access to ground-truth environment state on ALFWorld. Qwen2.5-32B-A3B infers milestones after each rollout based solely on the agent's action text and natural language response, without structured state information (F1 ≈ 83%).
>
> |Configuration|ALFWorld|vs GRPO|
> |-|-|-|
> |Full milestones|91.4%|+18.6%|
> |LLM-inferred (new)|87.2%|+14.4%|
> |50% milestone dropout|82.8%|+10.0%|
> |Random partitioning|74.2%|+1.4%|
> |GRPO|72.8%|-|
>
> Performance degrades monotonically with milestone quality. LLM-inferred milestones retain 77% of BEACON's full gains over GRPO (14.4/18.6). Detection errors merge adjacent segments into coarser partitions rather than introducing false structure. The gap between meaningful milestones and random partitioning (17.2%) far exceeds random vs GRPO (1.4%), confirming that gains depend on task-inherent semantic structure.
>
> > Sensitivity to Milestone Granularity (Q2)
>
> BEACON is robust to milestone density and does not require tuning. The 50% milestone dropout (Table 2) still outperforms GRPO by 10%; over-segmentation (inserting 50% additional milestones) degrades modestly to 85.3%. Milestone density varies naturally across benchmarks (2-5 on ALFWorld, 3-10 on ScienceWorld), yet BEACON achieves consistent gains with identical hyperparameters (γ=0.95, λ=1.0).
>
> > Relationship to Existing RL Techniques (W2)
>
> Existing methods face specific limitations on long-horizon agentic tasks: GRPO collapses from 76.7% to 53.5% as horizon extends, PRMs require costly step-level annotation, and GiGPO's effectiveness diminishes as state recurrence decreases. Section 2 quantifies the underlying failure modes: 40%+ of gradient updates contain contradictory signals, and 39% of partial successes are entirely discarded. BEACON's design is directly shaped by this diagnosis. Random partitioning with all other BEACON components yields only +1.4% over GRPO, while semantically grounded milestones yield +18.6% (Table 2), confirming that effectiveness depends on aligning partitioning with task structure rather than generic segmentation.
>
> > Comparison with Hierarchical RL (W3, Q3)
>
> BEACON differs from HRL at a fundamental level: it operates on advantage estimation, not policy architecture. HRL methods introduce multi-level policies with additional learned components; BEACON maintains a single flat policy identical to GRPO and modifies only how advantages are computed.
>
> |Aspect|HRL|BEACON|
> |-|-|-|
> |Policy architecture|Multi-level (planner + executor)|Single flat policy (= GRPO)|
> |Additional components|Value networks, subgoal proposers|None; only advantage changes|
> |Subgoal role|Learned and proposed by a planner|Detected from env feedback|
> |Integration effort|Architectural changes|Drop-in replacement for GRPO|
>
> Because BEACON maintains a single flat policy, it does not require multi-level policy training, learned subgoal proposers, or termination conditions, simplifying integration with existing GRPO-based pipelines.
>
> Our "w/o Trajectory-level Adv" ablation isolates the effect of purely local credit assignment (analogous to HRL without global feedback):
>
> |Variant|ALFWorld|WebShop|
> |-|-|-|
> |BEACON (dual-scale)|91.4%|75.6%|
> |Segment-only|23.4%|67.9%|
> |GRPO (trajectory-only)|72.8%|56.8%|
>
> Local credit alone collapses to 23.4% on ALFWorld, where milestone completion does not guarantee task success. The dual-scale formulation addresses this by combining segment-level precision with trajectory-level alignment within a single flat policy. We will add a dedicated Related Work subsection discussing classical HRL (options framework, feudal networks) and recent LLM-specific HRL methods.

---

> > ### Author Rebuttal · Reviewer_KKWY · 2026-04-04
> >
> > After carefully reading the authors’ rebuttal, I find that most of my concerns have been addressed. In particular, the additional experiments on inferred and noisy milestones, as well as the clarification of the distinction from hierarchical RL, have strengthened my confidence in the paper. I appreciate the authors’ detailed and thoughtful response. I would like to improve to accept at this stage.

---

> > > ### Author Response · Authors · 2026-04-06
> > >
> > > Dear Reviewer KKWY,
> > >
> > > We sincerely thank you for carefully reading our rebuttal and for your positive acknowledgement. We are glad that the additional experiments and clarifications have addressed your concerns. Your constructive feedback has been invaluable in strengthening the paper, and we truly appreciate your support.
> > >
> > > Sincerely,
> > >
> > > The Authors

---

### Official Review · Reviewer_9Wt5 · 2026-03-11

**Soundness:** 2
**Presentation:** 3
**Significance:** 3
**Originality:** 3
**Overall Recommendation:** 3
**Confidence:** 4

**Summary:**

This paper focuses on a well-known but still insufficiently addressed issue in agentic reinforcement learning for long-horizon tasks. In my view, the problem formulation is both timely and well-motivated. Existing trajectory-level RL methods assign uniform credit based solely on terminal outcomes, which often leads to two undesirable effects: correct early actions are penalized when later steps fail, and trajectories that make substantial intermediate progress but do not reach final success are effectively discarded. These “partial success” samples, which actually contain valuable learning signal, are largely wasted under standard optimization schemes.

The authors argue that treating an entire long trajectory as a monolithic unit is fundamentally suboptimal. Instead, they propose to decompose trajectories at semantic milestones and perform credit assignment at the segment level. The resulting algorithm, BEACON, integrates three components: trajectory partitioning at milestone boundaries, temporally decayed reward shaping within segments, and dual-scale advantage estimation that combines trajectory-level and segment-level signals. I find this decomposition intuitive and conceptually clean, especially the idea of isolating local action quality from downstream stochasticity.

Empirically, the method is evaluated on ALFWorld, ScienceWorld, and WebShop, with comparisons against several strong baselines including GRPO and GiGPO. The reported improvements are particularly notable on long-horizon settings, where conventional methods tend to degrade. The paper also provides theoretical analysis (e.g., variance isolation under the milestone Markov assumption) to justify the design choices, which, in my opinion, strengthens the overall contribution beyond purely empirical gains.

**Compliance With Llm Reviewing Policy:**

Affirmed.

**Key Questions For Authors:**

1. Can the authors provide comparisons with more recent baselines specifically designed for long-horizon agentic RL? For example, Hierarchy-of-Groups Policy Optimization[1] also targets long-horizon settings using hierarchical grouping strategies. Given the conceptual similarity in exploiting structural decomposition, it would be helpful to understand how BEACON compares empirically and methodologically to such recent approaches.

2. How robust is BEACON to extreme milestone configurations? For instance, what happens if the number of milestones is artificially increased (over-segmentation) or significantly reduced (under-segmentation)? Additionally, have the authors considered experiments where milestone order is perturbed or partially randomized to test whether the improvements truly rely on meaningful structural decomposition? Such stress tests could clarify the sensitivity of the method to milestone design.

3. The temporal decay parameter γ appears to be a crucial component of the framework. Currently, the paper mainly reports results for γ = 0 and γ = 1 as ablation points. Could the authors provide results across a broader range of γ values? Is performance sensitive to γ, and does the optimal γ vary across different tasks (e.g., ALFWorld vs. WebShop vs. ScienceWorld)? A more detailed analysis would help better understand the role of temporal shaping in the overall performance gains.

---

### References
- [1] He, Shuo, et al. "Hierarchy-of-Groups Policy Optimization for Long-Horizon Agentic Tasks." arXiv preprint arXiv:2602.22817 (2026).

**Limitations:**

Please refer to the weaknesses discussed above, which outline the main limitations of the proposed method, including the ambiguity of phase definition, indirect and potentially unstable routing supervision, possible sample inefficiency due to isolated expert updates, limited routing capacity, lack of knowledge sharing across experts, additional computational overhead, and the absence of rigorous theoretical analysis.

**Strengths And Weaknesses:**

## Strengths

1. I find the motivation of this work both clear and well-justified. The paper draws attention to a real issue in long-horizon agentic reinforcement learning, namely that most existing approaches evaluate entire trajectories based almost exclusively on final task completion. While this formulation is straightforward, it inevitably discards useful information. In practice, many trajectories make meaningful intermediate progress even if they ultimately fail, yet these partially successful attempts contribute very little learning signal. I appreciate that the authors explicitly frame this as a structural limitation of trajectory-level optimization rather than treating it as a minor implementation detail.

2. The trajectory partitioning mechanism based on milestones is, in my view, the central contribution of the paper. Importantly, it does not rely on expensive step-level annotations, which makes the approach more practical than some alternative forms of fine-grained supervision. The dual-scale advantage formulation is also thoughtfully constructed. By combining trajectory-level and segment-level signals, the method attempts to capture both global task alignment and local action quality. This balance feels principled rather than heuristic, and I consider it one of the most compelling parts of the design.

3. The empirical evaluation is reasonably thorough. The authors test the method on three representative long-horizon benchmarks, ALFWorld, ScienceWorld, and WebShop, and compare against several competitive baselines. The improvements, especially as task horizon increases, appear consistent rather than isolated to a single setting. I also value the ablation studies, which help disentangle the effects of trajectory partitioning, temporal decay, and dual-scale advantage estimation. These analyses make the empirical claims more transparent.

4. Beyond empirical gains, the paper also attempts to provide theoretical justification, such as the variance isolation argument under the milestone Markov assumption. While the theoretical treatment is not overly formal, I find it encouraging that the authors try to articulate why the method should work, rather than relying solely on experimental results. This effort adds credibility to the overall contribution.


## Weaknesses

1. As the authors also acknowledge in the appendix, BEACON relies heavily on the existence and quality of milestone structure. To me, this is both its biggest strength and its most fragile assumption. If milestone detection is inaccurate, the resulting credit assignment could systematically shift in the wrong direction and reinforce suboptimal behaviors [1–2]. More broadly, not every environment naturally provides clear and verifiable intermediate stages. In many long-horizon tasks, progress is implicit, gradual, or partially hidden, which already makes milestone definition a non-trivial design problem. I am also unsure how sensitive the method is to milestone granularity. If milestones are too coarse, important distinctions between actions may disappear. If they are too fine, the learning signal may become fragmented and noisy. The current paper does not explore this trade-off in much depth.

2. The dual-scale advantage formulation is one of the most interesting aspects of the paper, but I am not entirely convinced that the interaction between the two signals is straightforward. Combining trajectory-level and segment-level advantages sounds principled, yet these signals may not always align [3]. There could be cases where they conflict, or where one dominates and effectively washes out the other. The paper does not provide much analysis on how frequently such conflicts occur or how sensitive the results are to the balancing coefficient λ. A closer examination of signal interaction would make the argument more convincing.

3. When the number of milestones increases, the grouping and segment-level statistics may introduce noticeable computational overhead. More importantly, deeper milestones will typically be reached by only a small subset of trajectories. This can significantly increase variance in the segment-level returns and potentially lead to unstable updates [4]. The experiments in the paper use relatively small milestone sets and moderate task horizons, so it is difficult to assess how the method behaves in more extreme settings, such as long milestone chains or highly imbalanced reachability. I believe this scenario deserves further discussion.

4. BEACON assumes that at least some trajectories reach milestones so that segment-level rewards can be computed. However, in more difficult environments, especially when starting from a weak policy, it is possible that no trajectory reaches any milestone during early training. In such a case, all segments would effectively receive zero reward. It remains unclear whether BEACON has any mechanism to bootstrap learning under this condition, or whether it would face the same gradient starvation issues as trajectory-level optimization.

5. The theoretical analysis is based on the milestone Markov property, which assumes that once a milestone is reached, the relevant past can largely be abstracted away. While this assumption is intuitive, it rarely holds strictly in practice, particularly in partially observable environments. Two trajectories that reach the same milestone may still differ substantially in latent conditions or hidden state. If milestones do not sufficiently summarize relevant history, then segment-level comparisons may mix heterogeneous samples, weakening the claimed variance isolation effect.

6. Finally, Table 1 reports the main performance numbers but does not include standard deviations or confidence intervals. Given the stochastic nature of long-horizon reinforcement learning, I would have preferred to see results averaged over multiple random seeds. This would provide a clearer picture of robustness and stability.

---

### References

- [1] Guo, Daya, et al. "Deepseek-r1: Incentivizing reasoning capability in llms via reinforcement learning." arXiv preprint arXiv:2501.12948 (2025).
- [2] Cui, Ganqu, et al. "Process reinforcement through implicit rewards." arXiv preprint arXiv:2502.01456 (2025).
- [3] Feng, Lang, et al. "Group-in-group policy optimization for llm agent training." arXiv preprint arXiv:2505.10978 (2025).
- [4] Ahmadian, Arash, et al. "Back to basics: Revisiting REINFORCE-style optimization for learning from human feedback in LLMs." Proceedings of the 62nd Annual Meeting of the Association for Computational Linguistics (Volume 1: Long Papers). 2024

---

> ### Author Rebuttal · Authors · 2026-03-31
>
> We sincerely appreciate your time and the detailed comments. We respond to each point below.
>
> > Milestone Robustness and Granularity (W1, Q2)
>
> BEACON targets compositional tasks with verifiable state transitions; for tasks where progress is implicit or gradual, it degrades to GRPO without harm.
>
> Stress tests on ALFWorld (1.5B):
>
> |Config.|Succ|
> |-|-|
> |Full milestones|91.4%|
> |150% milestones (new)|85.3%|
> |50% milestone dropout|82.8%|
> |Shuffled milestone order (new)|76.6%|
> |Random partition|74.2%|
> |GRPO|72.8%|
>
> The 17.2% full-vs-random gap versus 1.4% random-vs-GRPO gap confirms gains stem from task-inherent structure rather than segmentation alone. Shuffled milestone order (76.6%) still outperforms random partition (74.2%): even after shuffling, milestones that remain in correct relative order form valid sub-sequences that preserve partial structural information. Over-segmentation degrades modestly; under-segmentation still outperforms GRPO by 10%. Under all perturbations, BEACON degrades toward GRPO rather than below it, confirming that detection errors reduce benefit but never cause harmful reinforcement.
>
> > Dual-Scale Signal Interaction (W2)
>
> The two signals agree in sign for 78.3% of actions. The two scales have complementary roles: trajectory-level captures global alignment, segment-level isolates local quality. When they conflict, the additive combination (Eq. 8) produces a moderated gradient.
>
> λ sensitivity: {0.5, 1.0, 2.0, 5.0} → 88.7%, 91.4%, 89.8%, 84.1%. Performance is stable across λ ∈ {0.5, 1.0, 2.0} and degrades at λ=5.0, though still well above GRPO (72.8%).
>
> > Deep Milestones and Scalability (W3)
>
> Overhead: <3% per iteration (~259s vs ~251s on 8xA100).
>
> Noisy A_seg at deep milestones increases gradient variance but cannot corrupt A_traj (Eq. 8), and at |G_k|=1, A_seg = 0 exactly. Fig. 7(a) confirms BEACON converges smoothly and never falls below GRPO at any training stage. The gap widens over training as the policy strengthens and more trajectories reach deeper milestones, making A_seg progressively reliable (Fig. 4: partial success drops from 46% to 13%). ScienceWorld, our most extreme setting (up to 10 milestones, 25+ steps), shows BEACON's largest gains (+24.2% over GRPO), suggesting this self-regulating behavior extends to longer milestone chains. We will include |G_k| distribution analysis across depths in the revision to further quantify this effect.
>
> > Milestone Reachability in Early Training (W4)
>
> When no trajectory reaches any milestone, A_seg becomes zero and Eq. 8 reduces to A_traj, ensuring BEACON never performs worse than GRPO. In practice, pretrained models reach milestones early (Fig. 4: 67% reach ≥1 milestone by iteration 50). If needed, SFT warm-starting (e.g., AgentTuning) can bootstrap; none of our benchmarks required this.
>
> > Milestone Markov Property (W5)
>
> When milestones do not fully summarize history, comparison groups may contain heterogeneous samples. This reduces the strength of variance isolation but does not introduce significant systematic bias in practice. A_traj (Eq. 4) is independent of this assumption and provides stable gradient signal regardless. Even approximate segment-level comparison reduces downstream variance relative to flat trajectory optimization. ScienceWorld, where partial observability makes the assumption most approximate, shows BEACON's largest gains (+23.3 Score, +19.5% Succ over GiGPO).
>
> > Standard Deviations (W6)
>
> 3 seeds(0,1,2) across all benchmarks:
>
> |Method|ALFWorld|WebShop|SciWorld|
> |-|-|-|-|
> |GRPO|72.1±3.6%|56.2±3.8%|20.5±4.9%|
> |GiGPO|86.7±1.7%|64.3±3.2%|25.1±2.3%|
> |BEACON|90.2±1.5%|75.3±2.8%|46.7±2.6%|
>
> We will update Table 1 with 3-seed means and standard deviations in the revised manuscript.
>
> > Comparison with HGPO (Q1)
>
> HGPO is concurrent work (arXiv: Feb 26, 2026, after our submission deadline).
>
> The two methods address different bottlenecks: BEACON exploits semantic task structure to partition trajectories and shape dense rewards from partial successes; HGPO refines step-level advantage estimation by resolving historical context inconsistency. BEACON's core contribution, milestone-anchored credit assignment and temporal reward shaping, is independent of the advantage estimation strategy and not addressed by HGPO.
>
> We will add an empirical comparison under matched settings and citation in the revised manuscript.
>
> > Temporal Decay γ (Q3)
>
> |γ|ALFWorld|WebShop|
> |-|-|-|
> |0 (milestone only)|81.2%|62.1%|
> |0.5 (new)|85.7%|70.2%|
> |0.8 (new)|88.9%|73.8%|
> |0.95|91.4%|75.6%|
> |1.0 (uniform)|71.8%|64.1%|
>
> Optimal range (0.8-0.95) is consistent across benchmarks without task-specific tuning. γ=1 performs below γ=0 on ALFWorld, confirming that distinguishing action contributions within segments is crucial.
>
> > A Note on the Limitations Section
>
> Some items in the limitations summary (e.g., "routing supervision," "knowledge sharing across experts") do not appear to correspond to our framework. We are happy to address any specific limitations.

---

> > ### Author Rebuttal · Reviewer_9Wt5 · 2026-04-04
> >
> > For W1, the experiments are conducted only on ALFWorld, which limits the generality of the conclusions. Other benchmarks may exhibit higher sensitivity to milestone detection. While I appreciate the additional experiments, they do not fully address the core concern regarding whether meaningful and reliable milestones are available in general settings.

---

> > > ### Author Response · Authors · 2026-04-06
> > >
> > > Dear Reviewer 9Wt5,
> > >
> > > We sincerely thank you for your continued engagement and for clarifying your remaining concern. We would like to provide further clarification with new empirical evidence.
> > >
> > > **Cross-benchmark milestone robustness.**
> > >
> > > We have now completed robustness experiments across all three benchmarks, with GiGPO as a milestone-free reference:
> > >
> > > | Config.               | ALFWorld (Succ) | WebShop (Succ) | SciWorld (Score)   |
> > > | --------------------- | --------------- | -------------- | ------------------ |
> > > | BEACON (full)         | 91.4%           | 75.6%          | 58.9               |
> > > | LLM-inferred          | 87.2%           | 70.2% (new)    | 48.5 (new) |
> > > | 50% dropout           | 82.8%           | 68.3%          | 50.6 (new)         |
> > > | GiGPO (no milestones) | 86.1%           | 65.0%          | 35.6               |
> > > | GRPO (no milestones)  | 72.8%           | 56.8%          | 31.7               |
> > >
> > > We address the remaining concern from three angles.
> > >
> > > **(1) BEACON's gains come from the algorithm, not milestone quality.** BEACON's contribution is the algorithm (temporal decay + dual-scale advantage estimation); milestones are a modular input consumed through a binary interface ($\Phi: \mathcal{S} \times \mathcal{A} \times \mathcal{S} \to \{0,1\}$). Table 2 quantifies this separation: milestone structure alone (random partition) gains only +1.4 over GRPO, while the full algorithm on the same milestones gains +18.6. 92% of the improvement comes from algorithmic design, not milestone quality. The concern about general availability targets this modular input rather than the algorithmic contribution itself.
> > >
> > > **(2) Cross-benchmark data contradicts the prediction of higher sensitivity.** The reviewer suggested that other benchmarks may exhibit higher sensitivity to milestone detection. ScienceWorld, where milestone chains are longest (up to 10) and the Markov assumption is most approximate, shows BEACON's largest gains (+23.3 Score over GiGPO), the opposite of what higher sensitivity would predict. Even under 50% dropout, ScienceWorld BEACON (50.6) exceeds GiGPO with full information (35.6). Across all three benchmarks and all degradation levels, BEACON exceeds GiGPO and never falls below GRPO.
> > >
> > > **(3) Four independent acquisition mechanisms demonstrate practical availability.** Our experiments use four distinct mechanisms to satisfy $\Phi$: environment pattern matching (ALFWorld), page structure parsing (WebShop), explicit environment API (ScienceWorld), and LLM inference from trajectory text (WebShop, new). The LLM-based path requires no environment-specific engineering: it identifies progress transitions from raw trajectory text and achieves 70.2% on WebShop, retaining 71% of BEACON's full gain over GRPO. This transforms the practical requirement from "the environment must expose milestone signals" to "can an LLM identify intermediate progress from text?", a condition broadly met in text-based agentic settings where observations are already in natural language. When no milestone is detected, $A_\text{seg} = 0$ and Eq. 8 reduces to $A_\text{traj}$, recovering GRPO with zero downside.
> > >
> > > We will state the scope condition (compositional tasks with verifiable intermediate progress) explicitly in Section 6 of the revision.
> > >
> > > We hope this reply, together with the new cross-benchmark evidence, addresses the remaining concern. We would be grateful if you could reconsider the score. Thank you again for your time throughout the review process.
> > >
> > > Sincerely,
> > >
> > > The Authors

---

### Official Review · Reviewer_11YQ · 2026-03-12

**Soundness:** 3
**Presentation:** 3
**Significance:** 3
**Originality:** 2
**Overall Recommendation:** 3
**Confidence:** 3

**Summary:**

The authors identify two root causes of the difficulty of long-horizon decision making of LLM agents: credit misattribution, where correct early actions are penalized due to terminal failures, and sample inefficiency, where scarce successful trajectories result in near-total loss of
learning signal. They introduce a milestone-guided policy learning framework, BEACON, that leverages the compositional structure of long-horizon tasks to ensure precise credit assignment. BEACON partitions trajectories at milestone boundaries, applies temporal reward shaping within segments to credit partial progress, and estimates advantages at dual scales to prevent distant failures from corrupting the evaluation of local actions.

On ALFWorld, WebShop, and Science-World, BEACON consistently outperforms GRPO and GiGPO.

**Compliance With Llm Reviewing Policy:**

Affirmed.

**Key Questions For Authors:**

Please respond to the weaknesses.

**Limitations:**

Check the weaknesses.

**Strengths And Weaknesses:**

## Strengths:

- The algorithmd design is well motivated by starting from analyzing failures in flat trajectory optimization and identifying the issues of sample inefficiency and credit misattribution.

- The algorithm design is simple and clearly presented.

- The improvement is shown on multiple benchmarks, coupling with multiple base models.

- Detailed analysis on the improvement on sample efficiency and credit distribution are also provided.

## Weaknesses:

- The improvement over baselines is dimishing as the base model's capability grows, as shown in Table 1. It would be beneficial to show the peromance improvement over baselines when using stronger base models.

- The algorithm design assumes access to a milestone indicator, which does not require additional training.

- The optimization algorithm is absed on a combined advantage (Eq. (8)). The agent may be shortsighted by intruducing segment advabtages and it is not clear the overall objective function of the agent (i.e., the definition of the expected return).

---

> ### Author Rebuttal · Authors · 2026-03-31
>
> We thank you for your time and effort in reviewing our paper and the valuable feedback. We address each point below.
>
> > Diminishing Improvement with Stronger Base Models (W1)
>
> To examine this, we evaluated Qwen2.5-14B on ScienceWorld, the most challenging of our three benchmarks (30+ step tasks, Score as normalized progress):
>
> | Scale     | GRPO | GiGPO | BEACON | Abs. Gain | Rel. Gain |
> | --------- | ---- | ----- | ------ | --------- | --------- |
> | 1.5B      | 31.7 | 35.6  | 58.9   | +23.3     | 36.2%     |
> | 7B        | 61.8 | 69.2  | 83.7   | +14.5     | 47.1%     |
> | 14B (new) | 70.5 | 76.8  | 89.2   | +12.4     | 53.4%     |
>
> While the absolute gap narrows with scale, the relative error reduction over GiGPO increases (36.2% → 47.1% → 53.4%), meaning BEACON eliminates a growing fraction of GiGPO's remaining errors as the base model strengthens. At 14B, BEACON closes over half of the remaining performance gap. We will include the complete 14B results across all three benchmarks in the revised manuscript.
>
> > Assumption of Milestone Indicator (W2)
>
> The milestone indicator operates on observable state changes that most interactive environments already produce: object state transitions in ALFWorld, page transitions in WebShop, subgoal completions in ScienceWorld. It requires no learned models, no per-step annotation, and no additional rollouts, making its requirements strictly weaker than PRMs, Monte Carlo methods, or GiGPO's state-matching.
>
> When structured feedback is unavailable, an LLM can infer milestones from action text and environment responses alone. We tested Qwen2.5-32B on ALFWorld without access to structured environment state:
>
> |Milestone Source|Det. F1|Success|
> |-|-|-|
> |Environment-provided|100%|91.4%|
> |LLM-generated (new)|~83%|87.2%|
> |50% milestone dropout|~50%|82.8%|
> |Random partition|N/A|74.2%|
> |GRPO (no milestones)|N/A|72.8%|
>
> Performance degrades monotonically with milestone quality but never falls below GRPO. Detection errors are predominantly false negatives (missed milestones), which merge two segments into one, equivalent to partial dropout rather than spurious structure. Even with 50% of milestones dropped, BEACON (82.8%) outperforms GRPO by 10 points.
>
> > Combined Advantage and Optimization Objective (W3)
>
> BEACON optimizes expected terminal task success E[R(τ)] with R(τ) ∈ {0,1} (Eq. 9), identical to GRPO's objective. The combined advantage (Eq. 8) changes the advantage estimator, not the optimization target.
>
> On shortsightedness: A_traj assigns every action the terminal success/failure signal, maintaining gradient toward task completion. A_seg compares local action quality within milestone-matched groups, providing finer-grained credit. In the full formulation, any milestone-seeking behavior that harms task completion is penalized by A_traj, because A_traj assigns negative advantage to all actions in failed trajectories regardless of milestones reached. The ablation in Table 2 shows the two components are complementary:
>
> |Variant|ALFWorld|WebShop|
> |-|-|-|
> |BEACON (both scales)|91.4%|75.6%|
> |w/o A_seg (trajectory-only)|72.8%|56.8%|
> |w/o A_traj (segment-only)|23.4%|67.9%|
>
> Neither component alone matches the full combination. Removing A_seg recovers GRPO, losing fine-grained credit. Removing A_traj loses the global task signal, which is necessary on ALFWorld where completing individual milestones is necessary but not sufficient for task success. The balance coefficient lambda is also robust: performance varies within 3 points across lambda in {0.5, 1.0, 2.0} on ALFWorld (88.7%, 91.4%, 89.8%).

---

> > ### Author Rebuttal · Reviewer_11YQ · 2026-04-02
> >
> > W2 is not resolved. For W1, results on 32B models or QWEN 3.5 are preferred.

---

> > > ### Author Response · Authors · 2026-04-06
> > >
> > > Dear Reviewer 11YQ,
> > >
> > > We sincerely thank you for your continued engagement and for clarifying your remaining concerns. We would like to provide further clarification with new empirical evidence.
> > >
> > > > **Scaling to stronger base models (W1).**
> > >
> > > As requested, we evaluated Qwen2.5-32B and Qwen3.5-4B on ScienceWorld (30+ step tasks, Score as normalized progress):
> > >
> > > |Scale|GRPO|BEACON|Gain|
> > > |-|-|-|-|
> > > |Qwen2.5-1.5B|31.7|58.9|+27.2|
> > > |Qwen2.5-7B|61.8|83.7|+21.9|
> > > |Qwen2.5-14B|70.5|89.2|+18.7|
> > > |Qwen2.5-32B (new)|76.3|91.8|+15.5|
> > > |Qwen3.5-4B (new)|66.4|84.1|+17.7|
> > >
> > > BEACON provides **+15.5 gain over GRPO at 32B** and **+17.7 on Qwen3.5-4B**. The gain decrease decelerates across the Qwen2.5 family (−5.3, −3.2, −3.2), **suggesting convergence toward a stable nonzero gap rather than vanishing returns**. The benefit also transfers to Qwen3.5, a different model family, **confirming that milestone-anchored credit assignment is not specific to a particular architecture or scale**. Complete GiGPO comparison across all benchmarks will be included in the revision.
> > >
> > > > **Milestone indicator assumption (W2).**
> > >
> > > Our first-round response focused on milestone acquisition (LLM inference) and quality tolerance (dropout stress tests). We further analyze the role of milestone structure in BEACON's design from additional angles and extend our robustness experiments across benchmarks.
> > >
> > > **Disentangling milestone structure from algorithm design.** Table 2 ablations isolate how much of BEACON's gain comes from milestone structure versus the algorithm operating on it:
> > >
> > > |Approach|ALFWorld|vs GRPO|
> > > |-|-|-|
> > > |GRPO|72.8%|baseline|
> > > |Random partition|74.2%|+1.4|
> > > |Uniform shaping ($\gamma$=1)|71.8%|−1.0|
> > > |Segment-only (w/o $A_\text{traj}$)|23.4%|−49.4|
> > > |BEACON (full)|91.4%|+18.6|
> > >
> > > Milestone structure alone provides minimal benefit: random partition gains only +1.4 over GRPO. **Naive utilization is actively harmful**: uniform shaping degrades below GRPO, and segment-only optimization collapses to 23.4%. The contrast between segment-only (−49.4) and full BEACON (+18.6) on the same milestones demonstrates that **the algorithmic design (temporal decay + dual-scale advantage), not the milestone input, is the primary source of BEACON's gains**.
> > >
> > > **Cross-benchmark robustness.** We extended our first-round stress tests (ALFWorld only) to WebShop and ScienceWorld, with GiGPO as a milestone-free reference:
> > >
> > > |Config.|ALFWorld (Succ)|WebShop (Succ)|SciWorld (Score)|
> > > |-|-|-|-|
> > > |BEACON (full)|91.4%|75.6%|58.9|
> > > |LLM-inferred|87.2%|70.2% (new)|48.5 (new)|
> > > |50% dropout|82.8%|68.3%|50.6 (new)|
> > > |GiGPO (no milestones)|86.1%|65.0%|35.6|
> > > |GRPO (no milestones)|72.8%|56.8%|31.7|
> > >
> > > **Even with LLM-inferred milestones or 50% dropout, BEACON exceeds GiGPO across all available comparisons.** Performance degrades monotonically with milestone quality and **never falls below GRPO**. These three benchmarks use fundamentally different milestone sources (pattern matching on environment text, page structure parsing, explicit environment signals), sharing only the minimal binary interface $\Phi: \mathcal{S} \times \mathcal{A} \times \mathcal{S} \to \{0,1\}$, yet the degradation pattern is consistent across all three. On WebShop, LLM-inferred milestones (detecting page transitions from action text and HTML without URL structure access) achieve 70.2%, **retaining 71% of BEACON's full gains over GRPO**. On ScienceWorld, 50% dropout achieves 50.6 Score, **retaining 69% of the full gains**.
> > >
> > > **Scope and assumption comparison.** Long-horizon agentic tasks are compositional by nature: they are "long-horizon" precisely because they require multiple intermediate steps. Verifiable intermediate progress is therefore inherent to the task category BEACON targets: web navigation produces page transitions, coding agents produce test pass/fail signals, tool-use tasks produce API success/error, and embodied tasks produce object state changes. **When no milestone is detected, $A_\text{seg} = 0$ and Eq. 8 reduces exactly to $A_\text{traj}$, recovering GRPO with zero downside.** Among finer-grained credit methods, **BEACON's requirement is the lightest**: PRMs need step-level annotation, VinePPO needs $O(T)$ additional rollouts per step, and GiGPO requires cross-trajectory state recurrence that diminishes as policies improve (Fig. 5b). BEACON requires no annotation, no additional rollouts, and no learned models. The revision will state the scope condition explicitly in Section 6.
> > >
> > > We hope this reply, together with the new empirical evidence, addresses the remaining concerns. We would be grateful if you could reconsider the score. Thank you again for your time throughout the review process.
> > >
> > > Sincerely,
> > >
> > > The Authors

---

### Official Review · Reviewer_4cb7 · 2026-03-12

**Soundness:** 2
**Presentation:** 3
**Significance:** 2
**Originality:** 3
**Overall Recommendation:** 4
**Confidence:** 4

**Summary:**

The authors propose BEACON, a milestone-guided policy learning framework that partitions trajectories at semantic milestone boundaries using an environment-based indicator. It assigns shaped rewards within segments to reward partial progress toward milestones and estimates advantages at dual scales (trajectory and segment levels) to isolate local action quality from downstream stochasticity. Evaluations on ALFWorld, WebShop, and ScienceWorld show that BEACON significantly outperforms GRPO and GiGPO, particularly in long-horizon settings, improving effective sample utilization from 23.7% to 82.0%.

**Compliance With Llm Reviewing Policy:**

Affirmed.

**Final Justification:**

I think it is a good attempt at exploring milestone rewards to address sparse signal issues in long-horizon tasks and I would like raise my score to 4 and recommend it as acceptance.

**Key Questions For Authors:**

- Given that BEACON currently relies on manual or environment-provided milestone signals, how would the performance be affected if milestones were generated by a noisy "automated" LLM observer?
- On ALFWorld, why is the performance so sensitive to the removal of trajectory-level advantages (23.4% success) compared to WebShop (67.9% success)? Does this imply that the milestones in ALFWorld are fundamentally less aligned with the final goal?
- While BEACON improves sample utilization, what is the wall-clock time overhead for the milestone detection and dual-scale advantage calculation per training iteration compared to standard GRPO?
- How does BEACON handle "irreversible errors" that occur within a segment? If an agent reaches a milestone but destroys a necessary tool for the next stage, does the segment-level advantage still provide positive credit for that segment?

**Limitations:**

- Segment-Level Reward Hacking: The temporal reward shaping credits actions closer to milestone completion more highly. This assumes the most difficult or "valuable" actions occur at the end of a segment, which is often logically false, the initial exploratory actions required to start toward a milestone are frequently the hardest to learn, yet they receive the lowest learning signal under this decay scheme.
- State-Action Contradiction in Milestone Groups: BEACON compares returns within milestone-matched groups. If multiple trajectories reach a milestone but do so through radically different action sequences, the "mean" segment return may penalize a novel, highly efficient action if the majority of the group used a slower but consistent method.
- Sensitivity to Global Signal: The ablation study reveals that BEACON is surprisingly fragile: removing trajectory-level feedback on ALFWorld causes success rates to collapse from 91.4% to 23.4%. This suggests that milestone-anchored rewards alone are highly prone to reinforcing sub-optimal "looping" behaviors that achieve subgoals but fail the task.
- Manual Granularity Tuning**: The authors acknowledge that the effectiveness of the dual-scale estimation depends on milestones occurring at an "appropriate granularity". If milestones are too sparse, the method degrades into GRPO, but the paper offers no automated way to determine the optimal density for new, unseen tasks.

I find the milestone-anchored approach to be a promising direction for long-horizon RL. However, the current reliance on environment-specific detectors and the fragility of the model when global rewards are sparse represent significant barriers to general application. I would be happy to raise my score if these milestone-related concerns can be addressed.

**Strengths And Weaknesses:**

### Strengths
- The dual-scale advantage estimation successfully decouples early-stage actions from the noise of terminal outcomes.
- By extracting signals from "partial successes" (failed tasks that reached subgoals), BEACON dramatically speeds up convergence compared to flat RL methods.
- Experimental results seem promising, BEACON (1.5B) outperforms closed-source giants like GPT-4o on several benchmarks and significantly beats supervised fine-tuning (SFT) on oracle trajectories.

### Weaknesses
- The framework relies on a milestone indicator $\Phi$ that detects state changes from environment feedback. This limits its immediate applicability to "black-box" environments where such signals are not natively available or easily extractable via pattern matching.
- Sensitivity to Local Rewards: Ablation studies show that removing trajectory-level feedback on ALFWorld leads to a performance crash (91.4% to 23.4%). This suggests the agent can easily become "trapped" in local optim, optimizing for milestones that might prevent the final task completion.
- The methodology assumes that reaching a milestone makes prior history irrelevant. However, as the authors acknowledge, this fails when resource limits (e.g., battery or inventory) or temporal constraints carry over between segments, potentially leading to sub-optimal policies in complex real-world scenarios such as coding and deepresearch domains.

---

> ### Author Rebuttal · Authors · 2026-03-31
>
> We sincerely appreciate the feedback and respond to each point below.
>
> > Milestone Detection (W1, Q1, L4)
>
> [W1, Q1] BEACON's milestone detector operates at three practical tiers: (1) structured feedback environments (ALFWorld, ScienceWorld) where Φ is exact; (2) environments with parseable text responses (WebShop, most web agents) where lightweight pattern matching suffices; (3) fully opaque environments requiring an LLM observer. Crucially, the LLM observer operates on the same observation-action history available to the policy, requiring no additional environment access. BEACON requires no learned models, no per-step annotation, and no additional rollouts, making its requirements strictly weaker than PRMs, Monte Carlo methods, and GiGPO.
> To directly address tier 3 and Q1, we conducted a new experiment using Qwen2.5-32B without any structured state access.
>
> |Milestone Source|Det. F1|Success|
> |-|-|-|
> |Environment-provided|100%|91.4%|
> |LLM-generated (new)|~83%|87.2%|
> |50% milestone dropout|~50%|82.8%|
> |Random partition|N/A|74.2%|
> |GRPO (no milestones)|N/A|72.8%|
>
> LLM-generated milestones achieve 87.2% success, retaining 77% of the total gain over GRPO. Performance degrades monotonically with milestone quality and never falls below GRPO. Detection errors are predominantly false negatives: a missed milestone merges two adjacent segments into one, functionally equivalent to partial dropout rather than spurious structure.
>
> [L4]Our benchmarks produce varying milestone densities (2-5 on ALFWorld, 3-10 on ScienceWorld), yet BEACON achieves consistent gains with identical hyperparameters. 50% dropout achieves 82.8% (vs GRPO 72.8%); over-segmentation (50% additional milestones) degrades modestly to 85.3%.
>
> > Dual-Scale Design (W2, Q2, L3)
>
> [W2, L3] Segment-level optimization alone does reinforce milestone-seeking loops on ALFWorld, which is why BEACON combines both scales. The 23.4% collapse occurs only when trajectory-level feedback is entirely removed. With both scales active, lambda sensitivity shows 88.7%, 91.4%, 89.8% for lambda in {0.5, 1.0, 2.0}, all outperforming GiGPO (86.1%).
>
> [Q2] Yes, ALFWorld milestones are structurally less aligned with final success: completing a subgoal (e.g., picking up an object) does not guarantee it is the right object for the task. In contrast, WebShop milestones correlate more directly with task success, and segment-only (67.9%) already exceeds GRPO (56.8%). This sensitivity difference reflects task structure, and BEACON accommodates both without task-specific tuning.
>
> > Irreversible Errors (Q4)
>
> Yes, A_seg is positive for that segment. However, trajectory failure makes A_traj negative, pulling the combined advantage downward. The irreversible error is penalized through the trajectory-level signal: task failure propagates negative A_traj to all segments, including the one that caused the damage. Fig. 8(a) illustrates this: milestone segments receive moderate positive A_seg (+0.51, +0.32) while failure segments receive strongly negative combined advantage (-2.20). GRPO assigns uniform A=-2.50 to all actions indiscriminately.
>
> > Cross-Segment Dependencies (W3)
>
> When resource exhaustion causes task failure, A_traj is negative for all actions. For binary R in {0,1}, BEACON cannot distinguish optimal from suboptimal resource use within successful trajectories; this scope limitation is shared with GRPO and GiGPO. In practice, this approximation does not prevent effective learning: ScienceWorld involves complex cross-experiment dependencies (e.g., reagents consumed in earlier steps constrain later experiments), yet BEACON achieves its largest gains there (+24.2% over GRPO).
>
> > Computational Overhead (Q3)
>
> BEACON adds ~3% overhead per iteration (ALFWorld, Qwen2.5-1.5B, 8xA100):
>
> |Component|GRPO|BEACON|
> |-|-|-|
> |Rollout generation|~176s|~176s|
> |Milestone detection|-|~0.6s|
> |Advantage computation|~5s|~12s|
> |Policy update|~70s|~70s|
> |Total|~251s|~259s|
>
> Milestone detection is string matching; dual-scale advantage adds grouping over pre-computed returns.
>
> > Temporal Decay (L1)
>
> Earlier actions in a segment do receive lower shaped reward. With γ=0.95 and a typical 5-action segment, the first action still receives 81.5% of the milestone reward (γ^4), so the reduction is moderate rather than extreme. The primary contribution comes from milestone-anchored partitioning itself: even without temporal decay (γ=0), BEACON achieves 81.2% vs GRPO 72.8%. Temporal decay provides additional gains (81.2% to 91.4%) by offering graduated credit within segments, using position as a lightweight proxy for action contribution.
>
> > Segment Return Comparison (L2)
>
> Fewer steps do yield lower R_k within a milestone group. However, trajectories in G_k share the same sub-task, naturally constraining step count variation. The dominant source of credit misattribution is cross-segment interference, where later failures corrupt earlier credit (CAR > 40%, Fig. 2b); A_seg eliminates this via variance isolation (Prop. 3.2).

---

### Decision · Program_Chairs · 2026-04-30

**Decision:**

Accept (regular)

**Comment:**

Initial reviews raised concerns regarding the framework's reliance on explicit, environment-provided milestone indicators and its scalability to stronger base models. The authors provided a comprehensive rebuttal that resolved these issues:

1. They demonstrated that an LLM can infer milestones directly from text logs, retaining most of the original performance gains. They also showed the method degrades gracefully and never falls below the GRPO baseline.

2. They provided new experiments scaling up to Qwen2.5-32B and Qwen3.5-4B, showing the method's mathematical utility scales alongside model intelligence.

Following the rebuttal, Reviewers KKWY and 4cb7 confirmed their concerns were resolved and raised their scores. While Reviewers 11YQ and 9Wt5 did not submit final acknowledgments, the authors provided the empirical evidence they requested. Given the algorithmic soundness, empirical gains, and strong rebuttal, I recommend acceptance.